# No-Regret is not enough! Bandits with General Constraints through Adaptive Regret Minimization

**Martino Bernasconi** [1]  **Matteo Castiglioni** [2]  **Andrea Celli** [1]

## Abstract

In the bandits with knapsacks framework (BwK) the learner has $m$ resource-consumption (*i.e.,* packing) constraints. We focus on the generalization of BwK in which the learner has a set of general long-term constraints. The goal of the learner is to maximize their cumulative reward, while at the same time achieving small cumulative constraints violations. In this scenario, there exist simple instances where conventional methods for BwK fail to yield sublinear violations of constraints. We show that it is possible to circumvent this issue by requiring the primal and dual algorithm to be *weakly adaptive*. Indeed, even without any information on the Slater's parameter $\rho$ characterizing the problem, the interaction between weakly adaptive primal and dual regret minimizers leads to a "self-bounding" behavior of dual variables. In particular, their norm remains suitably upper bounded across the entire time horizon even without explicit projection steps. By exploiting this property, we provide *best-of-both-worlds* guarantees for stochastic and adversarial inputs. In the first case, we show that the algorithm guarantees sublinear regret. In the latter case, we establish a tight competitive ratio of $\rho/(1 + \rho)$. In both settings, constraints violations are guaranteed to be sublinear in time. Finally, this results allow us to obtain new result for the problem of *contextual bandits with linear constraints*, providing the first no-$\alpha$-regret guarantees for adversarial contexts.

[1]Bocconi University, Milan, Italy [2]Politecnico di Milano, Milan, Italy. Correspondence to: Martino Bernasconi <martino.bernasconi@unibocconi.it>.

*Proceedings of the 42$^{nd}$ International Conference on Machine Learning*, Vancouver, Canada. PMLR 267, 2025. Copyright 2025 by the author(s).

## 1. Introduction

We consider a problem in which a decision maker tries to maximize their cumulative reward over a time horizon $T$, subject to a set of $m$ *long-term constraints*. At each round $t$, the learner chooses $x_t \in \mathcal{X}$ and, subsequently, observes a reward $f_t(x_t) \in [0, 1]$ and $m$ constraint functions $\boldsymbol{g}_t(x_t) \in [-1, 1]^m$. Then, the problem becomes that of finding a sequence of decisions which guarantees a reward close to that of the best fixed decision in hindsight, while satisfying long-term constraints $\sum_{t=1}^{T} \boldsymbol{g}_t(\boldsymbol{x}_t) \leq \boldsymbol{0}$ up to small sublinear violations. This framework subsumes the *bandits with knapsacks* (BwK) problem, where there are only resource-consumption constraints (Badanidiyuru et al., 2018; Agrawal & Devanur, 2019; Immorlica et al., 2022).

Inputs $(f_t, \boldsymbol{g}_t)$ may be either stochastic or adversarial. The goal is designing algorithms providing guarantees for both input models, without prior knowledge of the specific environment they will encounter. Achieving this goal involves addressing two crucial challenges which prevent a direct application of primal-dual approaches based on the LagrangeBwK framework in Immorlica et al. (2022).

### 1.1. Technical Challenges

In order to obtain meaningful regret guarantees, primal-dual frameworks based on LagrangeBwK need to control the magnitude of dual variables. This is necessary as dual variables appear in the loss function of the primal algorithm, and, therefore, influence the no-regret guarantees provided by the primal algorithm. In the context of knapsack constraints, this is usually achieved by exploiting the existence of a strictly feasible solution with Slater's parameter $\rho$, consisting of a *void action* which yields zero reward and resource consumption. For instance, the frameworks of (Balseiro et al., 2022; Castiglioni et al., 2022a) guarantee boundedness of dual multipliers through an explicit projection step on the interval $[0, 1/\rho]$. However, in settings with general constraints beyond resource consumption, it is often unreasonable to assume that the learner knows the Slater's parameter $\rho$ a priori. The problem of operating without knowledge of $\rho$ has been already addressed in the stochastic setting (Agrawal & Devanur, 2014; 2019; Yu et al., 2017; Wei et al., 2020; Castiglioni et al., 2022b). For instance, a

simple approach for the case of stochastic inputs involves adding an initial estimation phase to calculate an estimate of $\rho$, and subsequently treating this estimate as the true parameter (Castiglioni et al., 2022b). However, these techniques cannot be applied in adversarial environments as estimates of $\rho$ based on the initial rounds could be inaccurate about future inputs.

Primal-dual templates based on `LagrangeBwK` usually operate under the assumption that the primal and dual algorithms have the no-regret property. In the case of standard `BwK`, the no-regret requirement is sufficient to obtain optimal guarantees (see, *e.g.,* (Immorlica et al., 2022; Castiglioni et al., 2022a)). However, in our model, there exist simple instances in which the primal and dual algorithms satisfy the no-regret requirement, but the overall framework fails to guarantee small constraints violations (see Section 5.1). Moreover, known techniques to prevent this problem, such as introducing a *recovery phase* to prevent excessive violations, crucially require a priori knowledge of the Slater's parameter $\rho$ (Castiglioni et al., 2022b).

### 1.2. Contributions

Our approach is based on a generalization of the technique presented in Castiglioni et al. (2023) for online bidding under one budget and one return-on-investments constraint. The crux of the approach is requiring that both the primal and dual algorithms are *weakly adaptive*, that is, they guarantee a regret upper bound of $o(T)$ for each sub-interval of the time horizon (Hazan & Seshadhri, 2007). We generalize this approach to the case of $m$ general constraints, thereby providing the first primal-dual framework for this problem that can operate without any knowledge of Slater's parameter in both stochastic and adversarial environments.

First, we prove a "self-bounding" lemma for the case of $m$ arbitrary constraints. It shows that, if the primal and dual algorithms are weakly adaptive, then boundedness of dual multipliers emerges as a byproduct of the interaction between the primal and dual algorithm. Thus, it is possible to guarantee a suitable upper bound on the dual multipliers even without any information on Slater's parameter.

We use this result to prove *best-of-both-worlds* no-regret guarantees for primal-dual frameworks derived from `LagrangeBwK` which employ weakly adaptive primal and dual algorithms. Our guarantees will be modular with respect to the regret guarantees of the primal and dual algorithms. In presence of a suitable primal regret minimizer, we show that our framework yields the following no-regret guarantees while attaining sublinear constraints violations: in the stochastic setting, it guarantees sublinear regret with respect to the best fixed randomized strategy that is feasible in expectation. Remarkably, this result is obtained without having to allocate the initial $T^{1/2}$ rounds for estimating

the unknown parameter as in Castiglioni et al. (2022b). In the adversarial setting, our framework guarantees a competitive ratio of $\rho/(1 + \rho)$ against the best unconstrained strategy in hindsight. We provide a lower bound showing that this cannot be improved if constraint violations have to be $o(T)$. This is the first regret guarantee for our problem in adversarial environments.

Finally, we show that our model can be used to describe the *contextual bandits with linear constraints* (`CBwLC`) problem, which was recently studied by Slivkins et al. (2023b); Han et al. (2023) in the context of stochastic and nonstationary environments. Our framework allows to extend these works in two directions: we establish the first no-$\alpha$-regret guarantees for `CBwLC` when contexts are generated by an adversary, and we provide the first $\widetilde{O}(\sqrt{T})$ guarantees for the stochastic setting when the learner does not know an estimate of the Slater's parameter of the problem.

## 2. Related Work

**Bandits with Knapsacks.** The (stochastic) `BwK` problem was introduced an optimally solved by Badanidiyuru et al. (2013; 2018). Other algorithms with optimal regret guarantees have been proposed by Agrawal & Devanur (2014; 2019), whose approach is based on the paradigm of *optimism in the face of uncertainty*, and in (Immorlica et al., 2019; 2022). In the latter works, the authors propose the `LagrangeBwK` framework, which has a natural interpretation: arms can be thought of as primal variables, and resources as dual variables. The framework works by setting up a repeated two-player zero-sum game between a primal and a dual player, and by showing convergence to a Nash equilibrium of the expected Lagrangian game.

**Adversarial `BwK`.** The adversarial `BwK` problem was first introduced in Immorlica et al. (2019; 2022), where they studied the case in which the learner has $m$ knapsack constraints, and inputs are selected by an oblivious adversary. Their algorithm is based on a modified analysis of `LagrangeBwK`, and guarantees a $O(m \log T)$ competitive ratio. Subsequently, Kesselheim & Singla (2020) provided a new analysis obtaining a $O(\log m \log T)$ competitive ratio, which is optimal. In the case in which budgets are $\Omega(T)$, Castiglioni et al. (2022a) showed that it is possible to achieve a constant competitive ratio of $1/\rho$ where $\rho$ is the per-iteration budget.

**Beyond packing constraints.** Castiglioni et al. (2022a) studies a setting with general constraints analogous to ours, and show how to adapt the `LagrangeBwK` framework to obtain best-of-both-worlds guarantees when Slater's parameter is known a priori. Similar guarantees are also provided, in the stochastic setting, by Slivkins et al. (2023b), which then extend the results to the `CBwLC` model. Fi-

nally, the work of Castiglioni et al. (2023) introduces the use of weakly adaptive regret minimizers within the `LagrangeBwK` framework, and provides guarantees in the specific case of one budget constraint and one return-on-investments constraint.

**Contextual bandits (`CB`).** We briefly survey the most relevant works for our paper. Further references can be found in the monograph by Slivkins et al. (2019). As in (Slivkins et al., 2023a), we focus on `CB` with regression oracles (Foster et al., 2018; Foster & Rakhlin, 2020; Bietti et al., 2021; Simchi-Levi & Xu, 2022). The contextual version of `BwK` was first studied by Badanidiyuru et al. (2014) in the case of classification oracles. A regret-optimal and oracle-efficient algorithm for this problem was proposed by Agrawal et al. (2016) by exploiting the oracle-efficient algorithm for `CB` by Agarwal et al. (2014). The first regression-based approach for constrained `BwK` was proposed by Agrawal & Devanur (2016) by exploiting the optimistic approach for linear `CB` (Li et al., 2010; Chu et al., 2011; Abbasi-Yadkori et al., 2011). Han et al. (2023) propose a regression-based approach for a constrained `BwK` setup under stochastic inputs. Finally, a notable special case of constrained `CB` is online bidding under constraints (Balseiro & Gur, 2019; Celli et al., 2023; Gaitonde et al., 2023; Feng et al., 2023; Wang et al., 2023).

**Other related works.** Fikioris & Tardos (2023) show how to interpolate between the fully stochastic and the fully adversarial setting, depending on the magnitude of fluctuations in expected rewards and consumptions across rounds. Liu et al. (2022) study a non-stationary setting and provide no-regret guarantees against the best dynamic policy through a UCB-based algorithm. Some recent works explore the case in which resource consumptions in `BwK` can be non-monotonic (Kumar & Kleinberg, 2022; Bernasconi et al., 2023). Finally, a related line of works is the one on online allocation problems with fixed per-iteration budget, where the input pair of reward and costs is observed *before* the learner makes a decision (Balseiro et al., 2022; 2023).

## 3. Preliminaries

There are $T$ rounds and $m$ constraints. We denote with $\mathcal{X} \subset \mathbb{R}^K$ the decision space of the agent. At each round $t \in [\![T]\!]$, the agent selects an action $x_t \in \mathcal{X}$ and subsequently observes a reward $f_t(x_t)$ and costs function $\boldsymbol{g}_t(x_t) \in [-1, 1]^m$, with $f_t : \mathcal{X} \to [0, 1]$ and $g_{t,i} : \mathcal{X} \to [-1, 1]$ for each $i \in [\![m]\!]$.[1] The reward and cost functions can either be chosen by an oblivious adversary or drawn from a distribution. The goal of the decision maker is to max-

imize the cumulative reward $\texttt{Rew}(T) \coloneqq \sum_{t \in [\![T]\!]} f_t(x_t)$, while minimizing the cumulative violation $V_i(T)$ defined as

$$V_i(T) \coloneqq \sum_{t \in [\![T]\!]} g_{t,i}(x_t)$$

for each constraint $i \in [\![m]\!]$. We denote by $V(T) \coloneqq \max_{i \in [\![m]\!]} V_i(T)$ the maximum cumulative violation across the $m$ constraints.

### 3.1. Baselines

We will provide best-of-both-worlds no-regret guarantees for our algorithm, meaning that it achieves optimal theoretical guarantees both in the stochastic and adversarial setting. In this section, we introduce the baselines used to define the regret in these two scenarios.

**Adversarial Setting**   In the adversarial setting we employ the strongest baseline possible, *i.e.,* the best *unconstrained* strategy in hindsight:

$$\texttt{Opt}_{\texttt{Adv}} \coloneqq \sup_{x \in \mathcal{X}} \sum_{t \in [\![T]\!]} f_t(x).$$

This baseline is more powerful than the best fixed strategy which is feasible on average (Immorlica et al., 2022; Castiglioni et al., 2022a), which is the most common baseline in the literature. Our algorithm will yield an optimal competitive ratio against this stronger baseline. In this setting, we define $\rho_{\texttt{Adv}}$ as the feasibility parameter of the problem instance, *i.e.,* the largest reduction of cumulative violations that the agent is guaranteed to achieve by playing a "safe" strategy $\xi^\circ \in \Delta(\mathcal{X})$, where $\Delta(\mathcal{X})$ is the set of all probability measures on $\mathcal{X}$. Formally,

$$\rho_{\texttt{Adv}} \coloneqq - \max_{t \in [\![T]\!], i \in [\![m]\!]} \mathbb{E}_{x \sim \xi^\circ}[g_{t,i}(x)]$$

and

$$\xi^\circ \coloneqq \arg\inf_{\xi \in \Delta(\mathcal{X})} \max_{t \in [\![T]\!], i \in [\![m]\!]} \mathbb{E}_{x \sim \xi}[g_{t,i}(x)].$$

**Stochastic Setting**   When the reward and the costs are stochastic we denote by $\bar{f}$ and $\bar{\boldsymbol{g}}$ the mean of $f_t$ and $\boldsymbol{g}_t$, respectively. In particular, we have that the rewards are drawn so that $\mathbb{E}_{\texttt{Env}}[f_t(x)] = \bar{f}(x)$ (and similarly for the costs), where $\mathbb{E}_{\texttt{Env}}$ denotes expectation over the environment measure. We define the baseline for the stochastic setting as the best fixed *randomized* strategy that satisfies the constraints in expectation, which is the standard choice in Stochastic Bandits with Knapsacks settings (Badanidiyuru et al., 2013; Immorlica et al., 2022). Formally,

$$\texttt{Opt}_{\texttt{Stoc}} \coloneqq \sup_{\xi \in \Delta(\mathcal{X}): \, \mathbb{E}_{x \sim \xi}[\bar{\boldsymbol{g}}(x)] \leq \boldsymbol{0}} \mathbb{E}_{x \sim \xi}[\bar{f}(x)].$$

---

[1]In this work, for any $a, b \in \mathbb{N}$, with $a < b$ we denote with $[\![a]\!]$ the set $\{1, \ldots, a\}$ while $[\![a, b]\!]$ the set $\{a + 1, \ldots, b\}$.

Similarly to the adversarial case, we define the feasibility parameter $\rho_{\texttt{Stoc}}$ as the "most negative" cost achievable by randomized strategies *in expectation*:

$$\rho_{\texttt{Stoc}} := - \inf_{\xi \in \Delta(\mathcal{X})} \max_{i \in [\![m]\!]} \mathbb{E}_{x \sim \xi}[\bar{g}_i(x)].$$

As it is customary in relevant literature (see, *e.g.,* (Immorlica et al., 2022; Castiglioni et al., 2022a;b)), we make the following natural assumption about the existence of a strictly feasible solution. Note that we do not make any assumption on the variance of the samples $(f_t, \boldsymbol{g}_t)$ as we assume that they have bounded support, *i.e.,* with probability holds that $f_t(x) \in [0,1]$ and $g_{t,i}(x) \in [-1,1]$ for all $x \in \mathcal{X}$ and $i \in [\![m]\!]$.

**Assumption 3.1.** In the adversarial setting, the sequence of inputs $(f_t, \boldsymbol{g}_t)_{t=1}^T$ is such that $\rho_{\texttt{Adv}} > 0$. In the stochastic setting, the environment $\texttt{Env}$ is such that $\rho_{\texttt{Stoc}} > 0$.

**Remark 3.2.** We will describe a best-of-both-worlds type algorithm, that attains optimal guarantees both under stochastic and adversarial inputs, without knowledge of the specific setting in which the algorithm operates. It should be noted that $\rho_{\texttt{Adv}}$ and $\rho_{\texttt{Stoc}}$ are *not* known by the algorithm. While the algorithm could potentially efficiently estimate $\rho_{\texttt{Stoc}}$ in stochastic settings, as shown in Castiglioni et al. (2022b), acquiring knowledge of $\rho_{\texttt{Adv}}$ in the adversarial setting would necessitate information about future inputs. This requirement is generally unfeasible for most instances of interest.

## 4. On Best-Of-Both-Worlds Guarantees

We employ the expression *best-of-both-worlds* as defined in Balseiro et al. (2022) for the case of online allocation problems with resource-consumption constraints. In this context, we expect different types of guarantees depending on the input model being considered.

When inputs are stochastic, a best-of-both-worlds algorithm should guarantee that, given failure probability $\delta > 0$, with probability at least $1 - \delta$

$$\max(\texttt{Opt}_{\texttt{Stoc}} - \texttt{Rew}(T), V(T)) = \widetilde{O}(\sqrt{T}).$$

The dependency on $T$ is optimal since, in the worst case, it is optimal even without constraints (Auer et al., 2002).

In adversarial settings, a best-of-both-worlds algorithm should guarantee that, with probability at least $1 - \delta$,

$$\max \left( \texttt{Opt}_{\texttt{Adv}} - \alpha \texttt{Rew}(T), V(T) \right) = \widetilde{O}(\sqrt{T}),$$

where $\alpha > 1$ is the *competitive ratio*. In the $\texttt{BwK}$ scenario with only resource-consumption constraints, the optimal competitive ratio attainable is $\alpha = 1/\rho_{\texttt{Adv}}$. In that setting, $\rho_{\texttt{Adv}}$ denotes the per-iteration budget, which we can assume is equal for each resource without loss of generality.

---

**Algorithm 1** Primal-Dual Algorithm

1: **Input:** $\texttt{Alg}_{\texttt{P}}$ and $\texttt{Alg}_{\texttt{D}}$.
2: **for** $t = 1, 2, \ldots, T$ **do**
3:     **Primal decision:** $x_t \leftarrow \texttt{Alg}_{\texttt{P}}$
4:     **Dual decision:** $\boldsymbol{\lambda}_t \leftarrow \texttt{Alg}_{\texttt{D}}$
5:     **Observe:** $f_t(x_t)$ and $\boldsymbol{g}_t(x_t)$
6:     **Primal update:** feed $u_t^{\texttt{P}}(x_t)$ to $\texttt{Alg}_{\texttt{P}}$, where
7:        $u_t^{\texttt{P}}(x_t) \leftarrow f_t(x_t) - \langle \boldsymbol{\lambda}_t, \boldsymbol{g}_t(x_t) \rangle$
8:     **Dual update:**
8:        Feed $u_t^{\texttt{D}} : \boldsymbol{\lambda} \mapsto -f_t(x_t) + \langle \boldsymbol{\lambda}, \boldsymbol{g}_t(x_t) \rangle$ to $\texttt{Alg}_{\texttt{D}}$
9: **end for**

---

**Remark 4.1** (Comparison with $\texttt{BwK}$). To model $\texttt{BwK}$ problems with our framework, we define $g_t(x) = \frac{c_t^\top x - \rho_{\texttt{BwK}}}{1 - \rho_{\texttt{BwK}}}$, where $c_t \in [0,1]^m$ is the vector of costs of the $\texttt{BwK}$ instance and $\rho_{\texttt{BwK}} = B/T$. The zero cost action $a^\circ$ of the $\texttt{BwK}$ instance translates in the following $\rho_{\texttt{Adv}}$ of our framework:

$$g_t(a^\circ) = -\frac{\rho_{\texttt{BwK}}}{1 - \rho_{\texttt{BwK}}} = -\rho_{\texttt{Adv}}.$$

Inverting the formula we obtain $\rho_{\texttt{BwK}} = \frac{\rho_{\texttt{Adv}}}{1 + \rho_{\texttt{Adv}}}$ and $\frac{1}{\rho_{\texttt{BwK}}} = 1 + \frac{1}{\rho_{\texttt{Adv}}}$. This relation will be particularly helpful in interpreting our results and comparing it with previous works.

In our set-up, considering arbitrary and potentially negative constraints, we will present an algorithm for which the above holds for $\alpha := 1 + 1/\rho_{\texttt{Adv}}$. The following result shows that this competitive ratio is optimal. In particular, we show that it is not possible to obtain cumulative constraint violations of order $o(T)$ and competitive ratio strictly less that $1 + 1/\rho_{\texttt{Adv}}$ (omitted proofs can be found in the Appendix).

**Theorem 4.2** (Lower bound adversarial setting). *Consider the family of all adversarial instances with $\mathcal{X} = \{a_1, a_2\}$, each characterized by a parameter $\rho_{\texttt{Adv}}$ and optimal reward $\texttt{Opt}_{\texttt{Adv}}$. Then, no algorithm can achieve, on all instances, sublinear cumulative violations $\mathbb{E}[V(T)] = o(T)$ and $\texttt{Opt}_{\texttt{Adv}}/\mathbb{E}[\texttt{Rew}] < 1 + 1/\rho_{\texttt{Adv}}$.*

## 5. Lagrangian Framework

Given the reward function $f : \mathcal{X} \to [0,1]$ and the costs functions $\boldsymbol{g} : \mathcal{X} \to [-1,1]^m$ we define the Lagrangian $\mathcal{L}_{f,\boldsymbol{g}} : \mathcal{X} \times \mathbb{R}_+^m \to \mathbb{R}$ as:

$$\mathcal{L}_{f,\boldsymbol{g}}(x, \boldsymbol{\lambda}) := f(x) - \langle \boldsymbol{\lambda}, \boldsymbol{g}(x) \rangle.$$

We will consider a modular primal-dual approach that employs a *primal* algorithm $\texttt{Alg}_{\texttt{P}}$, producing primal decisions $x_t$, and a *dual* algorithm $\texttt{Alg}_{\texttt{D}}$ that produces dual decisions $\boldsymbol{\lambda}_t$ for all $t$. We assume that $\texttt{Alg}_{\texttt{P}}$ and $\texttt{Alg}_{\texttt{D}}$ produce their decisions in order to maximize their utilities $u_t^{\texttt{P}}$ and $u_t^{\texttt{D}}$, respectively. We define $u_t^{\texttt{P}} : x \mapsto \mathcal{L}_{f_t, \boldsymbol{g}_t}(x, \boldsymbol{\lambda}_t)$ and

$u_t^{\text{D}} : \boldsymbol{\lambda} \mapsto -\mathcal{L}_{f_t, \boldsymbol{g}_t}(x_t, \boldsymbol{\lambda})$. The regret of the primal algorithm $\texttt{Alg}_{\text{P}}$ on any subset $I \subseteq \llbracket T \rrbracket$ is defined as:

$$R_I^{\text{P}}(\mathcal{X}) := \sup_{x \in \mathcal{X}} \sum_{t \in I} [u_t^{\text{P}}(x) - u_t^{\text{P}}(x_t)].$$

The regret of the dual algorithm $\texttt{Alg}_{\text{D}}$ is defined similarly for any bounded subset $\mathcal{D} \subseteq \mathbb{R}_+$:

$$R_I^{\text{D}}(\mathcal{D}) := \sup_{\boldsymbol{\lambda} \in \mathcal{D}} \sum_{t \in I} [u_t^{\text{D}}(\boldsymbol{\lambda}) - u_t^{\text{D}}(\boldsymbol{\lambda}_t)].$$

For ease of notation we write $R_T^{\text{P}}(\mathcal{X})$ and $R_T^{\text{D}}(\mathcal{D})$ when $I = \llbracket T \rrbracket$, instead of $R_{\llbracket T \rrbracket}^{\text{P}}(\mathcal{X})$ and $R_{\llbracket T \rrbracket}^{\text{D}}(\mathcal{D})$.

The interaction of $\texttt{Alg}_{\text{P}}$ and $\texttt{Alg}_{\text{D}}$ with the environment is reported in Algorithm 1. Note that the feedback of $\texttt{Alg}_{\text{P}}$ is forced to be bandit by the fact that we do not have counterfactual information of $f_t$ and $\boldsymbol{g}_t$, however $\texttt{Alg}_{\text{D}}$ receives full feedback by design.

**Remark 5.1** (The Challenges of the Adversarial Setting). In the stochastic setting, adaptive regret minimization is not required (see, e.g., Slivkins et al. (2023b)), as one can directly analyze the expected zero-sum game between $\texttt{Alg}_{\text{P}}$ and $\texttt{Alg}_{\text{D}}$. However, in the adversarial setting, the algorithms $\texttt{Alg}_{\text{P}}$ and $\texttt{Alg}_{\text{D}}$ face a different zero-sum game at each time $t$. Indeed, since $f_t$ and $g_t$ are adversarial, the zero-sum game with payoffs $\mathcal{L}_{f_t, \boldsymbol{g}_t}(\cdot, \cdot)$ is only seen at time $t$. This is in contrast to what happens in the stochastic setting in which the zero-sum game $\mathcal{L}_{\bar{f}, \bar{g}}(\cdot, \cdot)$ at each time $t$ is the same for all time $t$.

### 5.1. No-Regret is Not Enough!

Typically, Lagrangian frameworks for constrained bandit problems are solved by instantiating $\texttt{Alg}_{\text{P}}$ and $\texttt{Alg}_{\text{D}}$ with two regret minimizers, which are algorithms guaranteeing $R_T^{\text{P}}(\mathcal{X}), R_T^{\text{D}}(\mathcal{D}) = o(T)$, respectively (Immorlica et al., 2022; Castiglioni et al., 2022a). The dual regret minimizer is usually instantiated with $\mathcal{D} := [0, M]^m$, for some constant $M > 0$. Ensuring that $\mathcal{D}$ is bounded is crucial to control the magnitude of primal utilities $u_t^{\text{P}}(\cdot)$, whose scale influences the magnitude of the primal regret. In the following example, we show that we cannot rely solely on arguments based on the *black-box* no-regret property of $\texttt{Alg}_{\text{P}}$ and $\texttt{Alg}_{\text{D}}$ and hence we need stronger guarantees then simple no-regret.

**Example 5.2.** *We have one constraint, i.e., $m = 1$ and the set $\mathcal{X} = \{a_1, a_2, a_3\}$ is a discrete set of 3 actions. The rewards of $a_1$ is always 0, i.e., $f_t(a_1) = 0$ for all $t \in \llbracket T \rrbracket$, while its cost is always $-\rho$, i.e., $g_{t,1}(a_1) = -\rho$ for all $t \in t$. The rewards for $a_2$ and $a_3$ are defined as follows: for $t \in \llbracket T/3 \rrbracket$ we have $f_t(a_2) = 0$ while $f_t(a_3) = 1$. On the other hand, for $t \in \llbracket T/3, 2T/3 \rrbracket$ we have $f_t(a_2) = 1$ while $f_t(a_3) = 0$. Finally $f_t(a_2) = f_t(a_3) = 0$ for all $t \in \llbracket 2T/3, T \rrbracket$. The costs for $a_2$ and $a_3$ are defined as*

*follows: for $t \in \llbracket 2T/3 \rrbracket$ we have $g_{t,1}(a_2) = g_{t,1}(a_3) = 0$, while $g_{t,1}(a_2) = g_{t,1}(a_3) = 1$ for all $t \in \llbracket 2T/3, T \rrbracket$. The instance is depicted in Figure 1.*

**Proposition 5.3.** *Consider the instance of Example 5.2. Even if $\texttt{Alg}_{\text{P}}$ and $\texttt{Alg}_{\text{D}}$ suffer regret less than or equal then zero, the primal-dual framework fails to achieve sublinear constraint violations.*

Intuitively, the reason for which a standard primal-dual framework fails in Example 5.2 is that the primal regret minimizer can accumulate enough negative regret in the first two phases to "absorb" large regret suffered in the third phase. This "laziness" of $\texttt{Alg}_{\text{P}}$ allows it to play actions in the last phase for which it incurs linear violations of the constraint. For more details see the proof of Proposition 5.3 in Appendix A. One could solve the problem employing the *recovery technique* proposed in Castiglioni et al. (2022b), which prescribes to minimize the violations at a prescribed time. However, selecting the right time to start the recovery phase crucially requires knowledge of the Slater's parameter, which is not available in our setting. The only approach which does not require knowledge of Slater's parameter is the one proposed in Castiglioni et al. (2023) for the case of *return-on-investment* constraints, whose core idea we describe in the next section.

**Remark 5.4.** We remark that it is not possible to prove that any choice of $\texttt{Alg}_{\text{P}}$ and $\texttt{Alg}_{\text{D}}$ satisfying the no-regret property fails in our setting. Indeed, we will end up choosing $\texttt{Alg}_{\text{P}}$ and $\texttt{Alg}_{\text{D}}$ algorithms that have a *stronger* no-regret property (and hence are also no-regret). Proposition 5.3 shows that our arguments and algorithms must necessarily rely on a stronger version of regret, specifically *no-adaptive regret*.

### 5.2. No-Adaptive Regret

The reason why generic regret minimizes fail to give satisfactory result on the instance described in Example 5.2 is that they fail to adapt to the changing environment, even if the regret of the primal is zero on the entire horizon $\llbracket T \rrbracket$, it fails to "adapt" in the final rounds $\llbracket 2T/3, T \rrbracket$. Indeed, in these last rounds, if the primal algorithm's objective is guaranteeing sublinear regret over $\llbracket T \rrbracket$, it is not required to updated its decision, since it accumulated large negative regret of $-2T/3$ regret in the initial rounds $\llbracket 2T/3 \rrbracket$. Therefore, standard no-regret guarantees are not enough.

A stronger requirement for the primal and dual algorithm is being *weakly adaptive* (Hazan & Seshadhri, 2007), that is, guaranteeing that in high probability $\sup_{I = \llbracket t_1, t_2 \rrbracket} R_I^{\text{P}, \text{D}} = o(T)$. Intuitively, this requirement would force $\texttt{Alg}_{\text{P}}$ to change its action during the last phase of Example 5.2. This idea was first proposed in (Castiglioni et al., 2023) for the specific case of a learner with one budget and one return-on-investments constraints. In the following section, we show

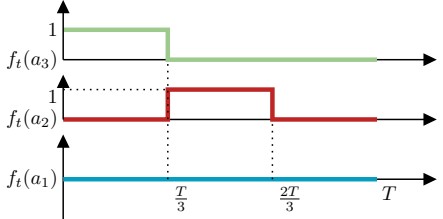 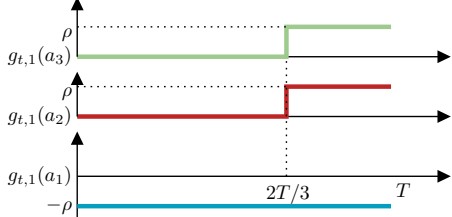

Figure 1: Reward and costs of each arm of the instance employed in Example 5.2.

how such approach can be extended to the case of general constraints.

# 6. Self-Bounding Lemma

One crucial difference with the previous literature is that the feasibility parameter is not known a priori, and thus we cannot directly bound the range of the Lagrange multipliers as in BwK. At a high level we want that, regardless of the choices of $f_t$ and $\boldsymbol{g}_t$, the $\ell_1$ norm of the Lagrange multipliers is bounded by a quantity that depends on the (unknown) parameters of the instance. However, for this to hold we need that the primal algorithm $\mathtt{Alg}_P$ is (almost) scale free, *i.e.,* that its regret scale quadratically in the unknown range of its reward function.[2] Formally:

**Definition 6.1.** For any $c \geq 1$, we say that $\mathtt{Alg}_P$ is a $c$-scale-free and weakly-adaptive regret minimizer if, for any subset of rounds $I = [\![t_1, t_2]\!] \subseteq [\![T]\!]$, with probability at least $1 - \delta$ it holds that

$$R_I^P(\mathcal{X}) \leq L^c \cdot \overline{R^P}_{T,\delta}(\mathcal{X}),$$

where the maximum module of the primal utilities is $\sup_{t\in[\![T]\!],x\in\mathcal{X}} |u_t^P(x)| =: L$, and $\overline{R^P}_{T,\delta}(\mathcal{X})$ depends only on $T$, $\delta$ and $\mathcal{X}$, and is non-decreasing in the length of the time horizon $T$.

Now, we show that *online gradient descent* (OGD) (Zinkevich, 2003) with a carefully defined learning rate yields the required self-bounding property both in the stochastic and adversarial setting.

**Lemma 6.2** (Self-bounding lemma). *Let* $\eta_{OGD} := \left(800 \cdot m \cdot \max\left\{\overline{R^P}_{T,\delta}(\mathcal{X}), E_{T,\delta}\right\}\right)^{-1}$, *then if* $\mathtt{Alg}_D$ *is* OGD *on the set* $\mathcal{D} = \mathbb{R}_{\geq 0}^m$, *and the primal algorithm* $\mathtt{Alg}_P$ *is* 2*-scale-free and has a high-probability weakly adaptive regret bound* $\overline{R^P}_{T,\delta}(\mathcal{X})$, *then with probability at least* $1 - \delta$:

$$\max_{t\in[\![T]\!]} \|\boldsymbol{\lambda}_t\|_1 \leq \frac{13m}{\rho},$$

*where* $\rho = \rho_{Adv}$ *or* $\rho = \rho_{Stoc}$ *depending on the setting and* $E_{T,\delta} := \sqrt{16T \log\left(2T/\delta\right)}$.

---
[2]Usually we say that an algorithm is scale-free (Orabona & Pál, 2018) if its regret scales linearly in the (unknown) range of its rewards, *i.e.,* 1-scale-free with our definition.

We remark that the self-bounding lemma shows that, if we choose OGD as $\mathtt{Alg}_P$, together with a carefully defined learning rate of

$$\eta_{OGD} = \widetilde{O}((m \max\{\overline{R^P}_{T,\delta}(\mathcal{X}), \sqrt{T}\})^{-1}),$$

then the $\ell_1$-norm of the variables $\boldsymbol{\lambda}_t$ is automatically bounded by the reciprocal of the feasibility parameter, even if the feasibility parameter is unknown to the learner. This is the central result that allows us to build algorithms that work without knowing Slater's parameter.

The proof relies on the key observation that Lagrangian multipliers jointly affect the primal utility but evolve independently. Thus, we need to reason about the joint behavior of the Lagrangian multipliers and look at their $\ell_1$ norm. In particular, the proof proceeds by contradiction (see Appendix B for the full proof): if the Lagrangian multipliers exceed a certain threshold, then they must have remained "large" for an extended period of time (Equation (3) in the Appendix). Then, we leverage the scale-free property of the primal regret minimizer, along with its regret bound relative to the feasible action that satisfies the constraints (Equation (4)), to establish that the cumulative primal utility is large over such interval (Equation (6)). However, this is in contradiction with the fact that the dual utility is also large (because of the growth of the Lagrangian multipliers, see Claim B.4). The proof of Claim B.4 is particularly involved, as it needs to analyze the separate behavior of the Lagrangian multipliers relative to different constraints. Indeed, it is possible for the norm of the multipliers to increase without having all individual components growing. This makes it nontrivial to conclude that the dual utility must have increased as well.

**Remark 6.3.** Even in the simplest instances of bandit problems one has $\overline{R^P}_{T,\delta}(\mathcal{X}) = \widetilde{\Omega}(\sqrt{T})$ and, therefore, we can assume that $\eta_{OGD} = \widetilde{O}\left((m\overline{R^P}_{T,\delta}(\mathcal{X}))^{-1}\right)$.

**Remark 6.4.** We will work with 2-scale-free algorithms, which suffice to obtain the desired guarantees for our framework. We observe that scale-free algorithms would yield a tighter bound of $1/\rho$ in the Theorems 7.2 and 7.3 and a simpler analysis of Lemma 6.2. However, scale-free algorithm are much more difficult to find and this would limit the extent to which our framework can be applied. On the other

hand, 2-scale-free algorithm seems to be more abundant (see, *e.g.,* Section 8). Indeed, as we show in Section 8, it is usually the case that setting the learning rate independent on the scale of the rewards provides 2-scale-freeness. We leave such characterization to future research.

## 7. General Guarantees

First, we exploit Lemma 6.2 to bound the total violations of the framework.

**Theorem 7.1.** *Let* $\mathrm{Alg}_D$ *be* $OGD$ *with learning rate* $\eta$ *as in Lemma 6.2, and let* $\mathrm{Alg}_P$ *any 2-scale-free algorithm with no-adaptive regret. Then, with probability at least* $1 - \delta$, *it holds that*

$$V_T = \widetilde{O}\left(\frac{m^2}{\rho}\overline{R^P}_{T,\delta}(\mathcal{X})\right),$$

*where* $\rho = \rho_{Adv}$ *in the adversarial setting and* $\rho = \rho_{Stoc}$ *in the stochastic.*

Moreover, the proof of Theorem 7.1 can be easily adapted to show that the violations of any constraint $i \in [\![m]\!]$ is bounded on any interval $[\![t]\!]$ with $t \in [\![T]\!]$.

Now, we prove that the framework, with high probability, yields optimal guarantees in both stochastic and adversarial settings. We start with the adversarial setting, for which the following result holds.

**Theorem 7.2.** *If* $\mathrm{Alg}_D$ *is* $OGD$ *with learning rate* $\eta_{OGD}$ *and domain* $\mathcal{D} := \mathbb{R}_{\geq 0}^m$, *and* $\mathrm{Alg}_P$ *is 2-scale-free, then, in the adversarial setting, with high probability:*

$$Rew \geq \frac{\rho_{Adv}}{1 + \rho_{Adv}}Opt_{Adv} - \widetilde{O}\left(\left(\frac{m}{\rho_{Adv}}\right)^2 \overline{R^P}_{T,\delta}(\mathcal{X})\right).$$

On the other hand, for the stochastic setting we can prove the following result:

**Theorem 7.3.** *If* $\mathrm{Alg}_D$ *is* $OGD$ *with learning rate* $\eta_{OGD}$ *and domain* $\mathcal{D} := \mathbb{R}_{\geq 0}^m$, *and* $\mathrm{Alg}_P$ *is 2-scale-free, then in the stochastic setting, in high probability:*

$$Rew \geq Opt_{Stoc} - \widetilde{O}\left(\left(\frac{m}{\rho_{Stoc}}\right)^2 \overline{R^P}_{T,\delta}(\mathcal{X})\right).$$

**Remark 7.4.** Any algorithm with vanishing constraints violations can be employed to handle also BwK constraints. In such setting, the learner has resource-consumption constraints with *hard stopping* (*i.e.,* once the budget for a resource is fully depleted the learner must play the void action until the end of time horizon). This does not yield any fundamental complication for our framework. Indeed, we could introduce an initial phase of $o(T)$ rounds in which the algorithm collects the extra budget needed to cover potential violations, before starting the primal-dual procedure.

## 8. Applications

In this section, we show how our framework can be instantiated to handle scenarios such as bandits with general constraints, as well as contextual bandits with constraints (*i.e.,* CBwLC). Thanks to the modularity of the results derived in the previous sections, we only need to provide an algorithm $\mathrm{Alg}_P$ which is 2-scale-free and weakly adaptive for a desired action space $\mathcal{X}$ and rewards $u_t^P$.

### 8.1. Bandits with General Constraints

In this setting, the action space is $\mathcal{X} = [\![K]\!]$. Castiglioni et al. (2023) showed that the EXP3-SIX algorithm introduced by Neu (2015) can be used as $\mathrm{Alg}_P$, since it guarantees sublinear weakly adaptive regret in high probability, and it is 2-scale-free.

**Theorem 8.1** (Theorem 8.1 of (Castiglioni et al., 2023))**.** *EXP3-SIX instantiated with suitable parameters guarantees that, with probability at least* $1 - \delta$ *that*

$$\sup_{I=[\![t_1,t_2]\!]} R_I^P(\mathcal{X}) = O\left(\sqrt{KT}\log\left(KT\delta^{-1}\right)\right).$$

Thus, by applying Theorem 7.1 on the violations, and Theorem 7.2 and Theorem 7.3 on the adversarial and stochastic reward guarantees respectively, we get the following result:

**Corollary 8.2.** *Consider a multi armed bandit problem with constraints. There exists an algorithm that w.h.p. guarantees, in the adversarial setting, violations at most* $\tilde{O}\left(\frac{m^2}{\rho_{Adv}}\sqrt{KT}\right)$ *and*

$$Rew \geq \frac{\rho_{Adv}}{1 + \rho_{Adv}}Opt_{Adv} - \tilde{O}\left(\frac{m^2}{\rho_{Adv}^2}\sqrt{KT}\right),$$

*while, in the stochastic setting, it guarantees violations at most* $\tilde{O}\left(\frac{m^2}{\rho_{Stoc}}\sqrt{KT}\right)$ *and reward at least*

$$Rew \geq Opt_{Stoc} - \tilde{O}\left(\frac{m^2}{\rho_{Stoc}^2}\sqrt{KT}\right).$$

### 8.2. Contextual Bandits with Constraints

Following Slivkins et al. (2023a), we apply our general framework to contextual bandits with regression oracles. In this setting, the decision maker observes a context $z_t \in \mathcal{Z}$ from some context set $\mathcal{Z}$, where $z_t$ is possibly chosen by an adversary. Then, the decision maker picks its decision $a_t$ from an action set $\mathcal{A}$. Then, the reward is computed as a function of the context and the action, *i.e.,* $f_t : \mathcal{Z} \times \mathcal{A} \to [0, 1]$, and similarly for the constraints $\boldsymbol{g}_t : \mathcal{Z} \times \mathcal{A} \to [-1, 1]^m$. At each $t$, $f_t$ and $\boldsymbol{g}_t$ are drawn from some distribution. More precisely, there exist a class $\mathcal{F}$ of functions and $\bar{f}, \bar{g}_i \in \mathcal{F}$ such that for all

**Algorithm 2** Primal-Dual Algorithm
for Contextual Bandits

1: **Input:** $\texttt{Alg}_\texttt{P}$ and $\texttt{Alg}_\texttt{D}$.
2: **for** $t = 1, 2, \ldots, T$ **do**
3:    Observe context $z_t$
4:    **Dual decision:** $\boldsymbol{\lambda}_t \leftarrow \texttt{Alg}_\texttt{D}$
5:    **Primal decision:**
6:       $a_t \leftarrow \texttt{Alg}_\texttt{P}(z_t, \boldsymbol{\lambda}_t)$
7:    **Observe:** $f_t(z_t, a_t)$ and $\boldsymbol{g}_t(z_t, a_t)$
8:    **Primal update:** feed $u_t^\texttt{P}(a_t)$ to $\texttt{Alg}_\texttt{P}$, where
9:       $u_t^\texttt{P}(a_t) = f_t(z_t, a_t) - \langle \boldsymbol{\lambda}_t, \boldsymbol{g}_t(z_t, a_t) \rangle$
10:    **Dual update:** feed $u_t^\texttt{D}$ to $\texttt{Alg}_\texttt{D}$, where
10:       $u_t^\texttt{D}(\boldsymbol{\lambda}) - f_t(z_t, a_t) + \langle \boldsymbol{\lambda}, \boldsymbol{g}_t(z_t, a_t) \rangle$
11: **end for**

---

**Algorithm 3** Primal Algorithm for Contextual Bandits

1: **Input:** Learning rate $\eta_\texttt{P}$
2: **Get regressors from online regression oracles:**
3:    $\hat{f}_t \leftarrow \mathcal{O}_f$, and $\hat{g}_{t,i} \leftarrow \mathcal{O}_i$ for all $i \in [\![m]\!]$
4: Observe context $z_t$ and dual variable $\boldsymbol{\lambda}_t$
5: For all $a \in \mathcal{A}$ compute $\hat{\mathcal{L}}_t(a) := \mathcal{L}_{\hat{f}_t, \hat{\boldsymbol{g}}_t}((z_t, a), \boldsymbol{\lambda}_t)$
6: Compute $\xi_t \in \Delta(\mathcal{A})$ as:

$$\xi_t(a) = \left( \mu_t + \eta_\texttt{P} \left( \max_{a'} \hat{\mathcal{L}}_t(a') - \hat{\mathcal{L}}_t(a) \right) \right)^{-1}$$

   ▷ *$\mu_t$ is such that $\xi_t \in \Delta(\mathcal{A})$*

7: Sample $a_t \sim \xi_t$ and return it.
8: **Update online regression oracles:**
9:    Feed $(z_t, a_t, f_t(z_t, a_t))$ to $\mathcal{O}_f$
10:    Feed $(z_t, a_t, g_{t,i}(z_t, a_t))$ to $\mathcal{O}_i \ \forall i \in [\![m]\!]$

---

$(z, a) \in \mathcal{Z} \times \mathcal{A}$ it holds that $\mathbb{E}[f_t(z, a) | z, a] = \bar{f}(z, a)$ and $\mathbb{E}[g_{t,i}(z, a) | z, a] = \bar{g}_i(z, a)$ for $i \in [\![m]\!]$.

We slightly modify the primal-dual algorithm to handle contexts. In particular, $\texttt{Alg}_\texttt{P}$ gets to observe a context $z_t$ before deciding their action. Formally, we can use the machinery introduced in Section 3 by taking $\mathcal{X}$ as the set of deterministic policies $\Pi := \{\pi : \mathcal{Z} \to \mathcal{A}\}$. Then, $u_t^\texttt{P}(\pi) = f_t(z_t, \pi(z_t)) - \langle \boldsymbol{\lambda}_t, \boldsymbol{g}_t(z_t, \pi(z_t)) \rangle$, and the action $a_t$ is computed through $\pi_t$ returned by the primal algorithm. Although this choice transforms the contextual framework into an application of the framework introduced in Section 3, in practical terms, it is simpler to think of $a_t$ as the direct output of $\texttt{Alg}_\texttt{P}$ upon observing the context $z_t$. The extended primal-dual framework is sketched in Algorithm 2.

We assume to have $m + 1$ online regression oracles $(\mathcal{O}_f, \mathcal{O}_1, \ldots, \mathcal{O}_m)$ for the functions $\bar{f}$ and $\bar{g}_1, \ldots, \bar{g}_m$, respectively. The regression oracle $\mathcal{O}_f$ produces, at each $t$, a regressor $\hat{f}_t \in \mathcal{F}$ that tries to approximate the *true* regressor $\bar{f}$. Then, the oracle is feed with a new data point, comprised of a context $z_t \in \mathcal{Z}$ and an action $a_t \in \mathcal{A}$,

and the performance of the regressor is evaluated on the basis of its prediction for the tuple $(z_t, a_t)$. The online regression oracle $\mathcal{O}_f$ is updated with the labeled data point $(z_t, a_t, f_t(z_t, a_t))$. Overall, its performance is measured by its cumulative $\ell_2$-error:

$$\texttt{Err}(\mathcal{O}_f) := \sum_{t \in [\![T]\!]} \left( \hat{f}_t(z_t, a_t) - \bar{f}(z_t, a_t) \right)^2 .$$

Each online regression oracle $(\mathcal{O}_i)_{i \in [\![m]\!]}$ works analogously, and its performance is measured by $\texttt{Err}(\mathcal{O}_i) := \sum_{t \in [\![T]\!]} (\hat{g}_i(z_t, a_t) - \bar{g}(z_t, a_t))^2$.

By combining the online regression oracles $\mathcal{O}_f$ and $\{\mathcal{O}_i\}_{i \in [\![m]\!]}$ we can build an online regression oracle $\mathcal{O}_\mathcal{L}$ for the Lagrangian which outputs regressors $\hat{\mathcal{L}}_t : \mathcal{Z} \times \mathcal{A} \to \mathbb{R}$ defined as:

$$\hat{\mathcal{L}}_t(z, a) = \mathcal{L}_{\hat{f}_t, \hat{\boldsymbol{g}}_t}((z, a), \boldsymbol{\lambda}_t) = \hat{f}_t((z, a)) - \langle \boldsymbol{\lambda}_t, \hat{\boldsymbol{g}}_t(z, a) \rangle,$$

while we define $\bar{\mathcal{L}}(z, a) := \mathcal{L}_{\bar{f}, \bar{\boldsymbol{g}}}((z, a), \boldsymbol{\lambda}_t)$. The $\ell_2$-error of $\mathcal{O}_\mathcal{L}$ can be bounded via the following extension of Theorem 16 in Slivkins et al. (2023b).

**Lemma 8.3.** *The error of $\mathcal{O}_\mathcal{L}$ can be bounded as*

$$\texttt{Err}(\mathcal{O}_\mathcal{L}) \leq 2\texttt{Err}(\mathcal{O}_f) + 2 \left( \sup_{t \in [\![T]\!]} \|\boldsymbol{\lambda}_t\|_1 \right)^2 \sum_{i \in [\![m]\!]} \texttt{Err}(\mathcal{O}_i).$$

The fundamental idea of (Foster & Rakhlin, 2020) is to reduce (unconstrained) contextual bandit problems to online linear regression. Recently, this ideas was extended in (Slivkins et al., 2023a; Han et al., 2023) in order to design a primal algorithm $\texttt{Alg}_\texttt{P}$ capable of handling stochastic contextual bandits with constraints (see Algorithm 3).

To apply Algorithm 3 to our framework we need to find an algorithm $\texttt{Alg}_\texttt{P}$ which is 2-scale-free and weakly adaptive with high probability. We extend the result of (Foster & Rakhlin, 2020) to prove that their reduction actually satisfies the required guarantees.

**Lemma 8.4.** *Assume that $\max\{\texttt{Err}(\mathcal{O}_f), \texttt{Err}(\mathcal{O}_i)\} \leq \overline{\texttt{Err}}$. Then, we have that Algorithm 3 with $\eta_\texttt{P} := \sqrt{KT}$ guarantees that*

$$\sup_{I = [\![t_1, t_2]\!]} R_I^\texttt{P}(\Pi) = \tilde{O} \left( m \cdot \overline{\texttt{Err}} \cdot L^2 \cdot \sqrt{KT} \right)$$

*with high probability, where $L := \sup_{t \in [\![T]\!], \pi \in \Pi} |u_t^\texttt{P}(\pi)|$.*

Equipped with a 2-scale free algorithm that suffers no adaptive regret with high probability, we can combine $\texttt{Alg}_\texttt{P}$ with the results of Theorems 7.1 to 7.3 to prove the first optimal guarantees for CBwLC with adversarial contexts.

**Corollary 8.5.** *Consider a functional class $\mathcal{F}$ and an online regression oracle that guarantees $\ell_2$-error $\overline{\texttt{Err}}$. There exists an algorithm that w.h.p. guarantees violations at most $\tilde{O}\left(\frac{m^3}{\rho_{Adv}}\overline{\texttt{Err}}\sqrt{KT}\right)$ and reward at least*

$$\texttt{Rew} \geq \frac{\rho_{Adv}}{1+\rho_{Adv}}\texttt{Opt}_{Adv} - \tilde{O}\left(\overline{\texttt{Err}}\frac{m^3}{\rho_{Adv}^2}\sqrt{KT}\right)$$

*in the adversarial setting, while it guarantees violations at most $\tilde{O}\left(\frac{m^3}{\rho_{Stoc}}\overline{\texttt{Err}}\sqrt{KT}\right)$ and reward at least*

$$\texttt{Rew} \geq \texttt{Opt}_{Stoc} - \tilde{O}\left(\overline{\texttt{Err}}\frac{m^3}{\rho_{Stoc}^2}\sqrt{KT}\right)$$

*in the stochastic setting.*

Foster & Rakhlin (2020) includes many examples of functional classes $\mathcal{F}$ that have good online regression oracles, meaning that their error is subpolynomial in the time horizon $T$. We report here some notable mentions for completeness.

If $\mathcal{F}$ is a finite set of functions we have that $\overline{\texttt{Err}} = O(\log|\mathcal{F}|)$, which comes from using as regression oracles the Vovk forecaster (Vovk, 1995). Another important examples is the case in which $\mathcal{F}$ is the class of linear functions, *i.e.*, $\mathcal{F} = \{h(z,a) = \langle z_a, \theta\rangle : \theta \in \mathbb{R}^d, \|\theta\|_2 \leq 1\}$, *i.e.*, each actions $a$ is associated with a known feature vector $z_a \in \mathbb{R}^d$ which generates the reward/costs trough a unknown parameter $\theta$ that characterize the linear function. Here, there exists a online regression oracle which provides $\ell_2$-error $\overline{\texttt{Err}} = O(d\log(T/d))$ (Azoury & Warmuth, 2001).

## Acknowledgements

Funded by the European Union. Views and opinions expressed are however those of the author(s) only and do not necessarily reflect those of the European Union or the European Research Council Executive Agency. Neither the European Union nor the granting authority can be held responsible for them.

This work is supported by ERC grant (Project 101165466 — PLA-STEER), by the FAIR (Future Artificial Intelligence Research) project, funded by the NextGenerationEU program within the PNRRPE-AI scheme (M4C2, Investment 1.3, Line on Artificial Intelligence), and by the EU Horizon project ELIAS (European Lighthouse of AI for Sustainability, No. 101120237).

## Impact Statement

This paper presents work whose goal is to advance the field of Machine Learning. There are many potential societal consequences of our work, none which we feel must be specifically highlighted here.

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

## A. Omitted Proofs from Section 4 and Section 5

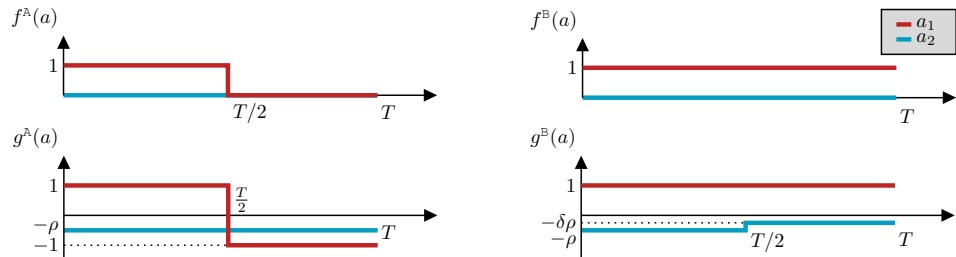

Figure 2: Lower bound adversarial setting: rewards and costs in the two instances A and B.

**Theorem 4.2** (Lower bound adversarial setting). *Consider the family of all adversarial instances with $\mathcal{X} = \{a_1, a_2\}$, each characterized by a parameter $\rho_{\text{Adv}}$ and optimal reward $\text{Opt}_{\text{Adv}}$. Then, no algorithm can achieve, on all instances, sublinear cumulative violations $\mathbb{E}[V(T)] = o(T)$ and $\text{Opt}_{\text{Adv}}/\mathbb{E}[\text{Rew}] < 1 + 1/\rho_{\text{Adv}}$.*

*Proof.* We show that, for all $\epsilon > 0$ and $\delta \in (0, 1)$, there exists two instances such that it is impossible to obtain $\mathbb{E}[V(T)] \leq \epsilon T$ and

$$\frac{\text{Opt}_{\text{Adv}}}{\mathbb{E}[\text{Rew}]} < \frac{1 + \rho_{\text{Adv}}}{\rho_{\text{Adv}}(1 + \delta) + 2\epsilon}$$

in both instances. The two instances are denoted by A and B respectively, with $\mathcal{X} = \{a_1, a_2\}$ and sequences of inputs of length $T$. The two instances are identical in the first $T/2$ rounds. Rewards in instance A are, for each $t \in [\![T]\!]$, $f_t^{\text{A}}(a_2) = 0$ and $f_t^{\text{A}}(a_1) = \mathbb{1}[t \leq T/2]$. On the other hand, in instance B we have $f_t^{\text{B}}(a_2) = 0$, and $f_t^{\text{B}}(a_1) = 1$ for all $t \in [\![T]\!]$. Costs for the first instance A are define as

$$g_t^{\text{A}}(a_1) := \begin{cases} 1 & \text{if } t \leq T/2 \\ -1 & \text{otherwise} \end{cases},$$

and $g_t^{\text{A}}(a_2) = -\rho$ for all $t \in [\![T]\!]$. In the second instance B, costs are $g_t^{\text{B}}(a_1) = 1$ for all $t \in [\![T]\!]$, and

$$g_t^{\text{A}}(a_2) := \begin{cases} -\rho & \text{if } t \leq T/2 \\ -\delta\rho & \text{otherwise} \end{cases},$$

for some $\delta > 0$. The two instances are depicted in Figure 2.

Let $N$ be the expected number of times that action $a_1$ is played in rounds $[\![T/2]\!]$, that is

$$N := \sum_{t \in [\![T/2]\!]} \mathbb{E}^{\text{A}}[x_t = a_1] = \sum_{t \in [\![T/2]\!]} \mathbb{E}^{\text{B}}[x_t = a_1],$$

where expectation is with respect to the algorithm's randomization. We observe that the algorithm plays in the same way in both instances up to time $T/2$, as they are identical (formally, the KL between instance A and B is zero in the first $T/2$ rounds). Then, we have that the optimal action in instance A is to play deterministically action $a_1$. Therefore, $\text{Opt}_{\text{Adv}}^{\text{A}} = T/2$. The expected reward in instance A comes only from the number of plays of $a_1$ in the first $T/2$ rounds: $\mathbb{E}^{\text{A}}[\text{Rew}] = N$. On the other hand, call $M$ the expected number of times an algorithm plays action $a_1$ in the last $[\![T/2, T]\!]$ rounds of instance B, that is

$$M := \sum_{t \in [\![T/2, T]\!]} \mathbb{E}^{\text{B}}[x_t = a_1].$$

We have that, in order to have $\mathbb{E}^{\text{B}}[V(T)] \leq \epsilon T$ violations in the second instance, we need to play $a_1$ a small number of times:

$$M - \delta\rho\left(\frac{T}{2} - M\right) + N - \rho\left(\frac{T}{2} - N\right) \leq \epsilon T,$$

which yields

$$N \leq \frac{T(\rho(\delta + 1) + 2\epsilon)}{2(\rho + 1)}.$$

Then, we get that

$$\frac{\texttt{Opt}_{\texttt{Adv}}^{\texttt{A}}}{\mathbb{E}^{\texttt{A}}[\texttt{Rew}]} \geq \frac{1+\rho}{\rho(1+\delta)+2\epsilon},$$

which concludes the proof since $\rho_{\texttt{Adv}}^{\texttt{A}} = \rho$. $\square$

**Proposition 5.3.** *Consider the instance of Example 5.2. Even if $\texttt{Alg}_{\texttt{P}}$ and $\texttt{Alg}_{\texttt{D}}$ suffer regret less than or equal then zero, the primal-dual framework fails to achieve sublinear constraint violations.*

*Proof.* Consider the instance described in Example 5.2, and consider an algorithm $\texttt{Alg}_{\texttt{P}}$ for $\mathcal{X} = \{a_1, a_2, a_3\}$ such that $x_t = a_3$ for $t \in [\![T/3]\!]$, while $x_t = a_2$ for $t \in [\![T/3, T]\!]$. Moreover, consider an algorithm $\texttt{Alg}_{\texttt{D}}$ instantiated on $\mathcal{D} = [0, M]$, with $M \geq 1/\rho$, that plays $\lambda_t = 0$ for all $t \in [\![2T/3]\!]$, and $\lambda_t = M$ for all $t \in [\![2T/3, T]\!]$.

We start by analyzing the primal regret achieved by $\texttt{Alg}_{\texttt{P}}$:

$$
\begin{aligned}
R_T^{\texttt{P}} &:= \sup_{x \in \mathcal{X}} \sum_{t \in [\![T]\!]} [f_t(x) - f_t(x_t) - \lambda_t(g_{t,1}(x) - g_{t,1}(x_t))] \\
&= \sup_{x \in \mathcal{X}} \sum_{t \in [\![T]\!]} [f_t(x) - \lambda_t g_{t,1}(x)] - \frac{2}{3}T + \frac{M\rho}{3}T \\
&= \sum_{t \in [\![T]\!]} [f_t(a_1) - \lambda_t g_{t,1}(a_1)] + \frac{T}{3}(M\rho - 2) \\
&= \rho M \frac{T}{3} + \frac{T}{3}(M\rho - 2) \\
&= \frac{T}{3}(2M\rho - 2) \leq 0,
\end{aligned}
$$

where we replaced the sup with the utility at $a_1$ since $M \geq 1/\rho$. Moreover, the dual regret is such that

$$
\begin{aligned}
R_T^{\texttt{D}} &:= \sup_{\lambda \in [0,M]} \sum_{t \in [\![2T/3, T]\!]} (\lambda - M)\, g_{t,1}(x_t) \\
&= \sup_{\lambda \in [0,M]} \frac{T}{3}(\lambda - M)\, \rho = 0.
\end{aligned}
$$

However, for a suitable choice of $\rho$, the violations are linear in $T$ since

$$V_1(T) := \sum_{t \in [\![T]\!]} g_{t,1}(x_t) = \frac{\rho}{3}T = \Omega(T).$$

This concludes the proof. $\square$

## B. Proof of Lemma 6.2

We start by providing the following auxiliary lemmas.

**Lemma B.1.** *Let $\boldsymbol{y}_t \in \mathbb{R}_{\geq 0}^m$ be generated by $\texttt{OGD}$ with learning rate $\eta$ and utilities $\boldsymbol{y} \mapsto \langle \boldsymbol{y}, \boldsymbol{g}_t \rangle$, where $\|\boldsymbol{g}_t\|_\infty \leq 1$ for all $t \in [\![T]\!]$. Then:*

$$\big| \|\boldsymbol{y}_{t+1}\|_1 - \|\boldsymbol{y}_t\|_1 \big| \leq m \cdot \eta$$

*Proof.* The update of the $i$-th component of $\boldsymbol{y}_{t+1}$ can be written as:

$$y_{t+1,i} := \max(0, y_{t,i} + \eta g_{t,i}).$$

If $g_{t,i} \geq 0$ then the update can be simplified to $y_{t+1,i} = y_t + \eta g_{t,i} \leq y_t + \eta$. If $g_{t,i} < 0$ then $y_{t+1,i} \geq y_{t,i} + \eta g_{t,i} \geq y_{t,i} - \eta$. Thus $|y_{t+1,i} - y_{t,i}| \leq \eta$ for all $i \in [\![m]\!]$. By summing over all component we have that $\|\boldsymbol{y}_{t+1} - \boldsymbol{y}_t\|_1 \leq m \cdot \eta$. By triangular inequality we have the desired statement. $\square$

**Lemma B.2.** *[Chapter 10 of Hazan (2016)] For any $t_1, t_2 \in [\![T]\!]$ with $t_1 < t_2$, it holds that if $\boldsymbol{\lambda}_t$ is generated by* OGD *with learning rate $\eta > 0$ on a set $\mathcal{D}$, then:*

$$R^p_{[\![t_1, t_2]\!]}(\{\boldsymbol{\lambda}\}) \leq \frac{\|\boldsymbol{\lambda} - \boldsymbol{\lambda}_{t_1}\|^2_2}{2\eta} + \frac{1}{2}\eta m T.$$

*with probability probability one on the randomization of the algorithm, i.e., $\delta = 0$. Moreover it also holds component-wise, i.e., for all $\lambda \geq 0$:*

$$\sum_{t \in [\![t_1, t_2]\!]} (\lambda - \lambda_t) g_t(x_t) \leq \frac{(\lambda - \lambda_{t_1})^2}{2\eta} + \frac{1}{2}\eta T.$$

**Lemma B.3.** *In the stochastic setting, for any $\xi \in \Delta(\mathcal{X})$ and $\delta \in (0, 1]$, with probability at least $1 - \delta$, it holds that:*

$$\sum_{t \in I} \mathbb{E}_{x \sim \xi} \left[ \langle \boldsymbol{\lambda}_t, \boldsymbol{g}_t(x) \rangle \right] \leq \sum_{t \in I} \mathbb{E}_{x \sim \xi} \left[ \langle \boldsymbol{\lambda}_t, \bar{\boldsymbol{g}}_t(x) \rangle \right] + M E_{T, \delta} \quad \text{and} \tag{1}$$

$$\sum_{t \in I} \mathbb{E}_{x \sim \xi} \left[ f_t(x) \right] \geq \sum_{t \in I} \mathbb{E}_{x \sim \xi} \left[ \bar{f}(x) \right] - E_{T, \delta}, \tag{2}$$

*for any interval $I = [t_1, t_2] \subseteq [T]$, where $E_{T, \delta} := \sqrt{16 T \log \left( \frac{2T}{\delta} \right)}$ and $M = \sup\limits_{t \in [\![T]\!]} \|\boldsymbol{\lambda}\|_1$.*

*Proof.* We start by proving that the all the inequalities of Equation (1) holds simultaneously with probability $1 - \delta/2$. We have that given a $I = [t_1, t_2] \subseteq [T]$, with probability at least $1 - \delta/(2T^2)$,

$$\sum_{t \in I} \mathbb{E}_{x \sim \xi} \left[ \langle \boldsymbol{\lambda}_t, \boldsymbol{g}_t(x) \rangle \right] - \sum_{t \in I} \mathbb{E}_{x \sim \xi} \left[ \langle \boldsymbol{\lambda}_t, \bar{\boldsymbol{g}}_t(x) \rangle \right] \leq M \sqrt{8|I| \log \left( \frac{2T^2}{\delta} \right)} \leq M \sqrt{16 T \log \left( \frac{2T}{\delta} \right)},$$

where the first inequality holds by Azuma-Hoeffding inequality. By taking a union bound over all possible intervals $I$ (which are at most $T^2$), we obtain that all the first set of equations holds with probability at least $1 - \delta/2$.

Equation (2) can be proved in a similar way. Indeed, for any fixed interval $I = [t_1, t_2] \subseteq [T]$, and for any strategy mixture $\xi \in \Delta(\mathcal{X})$, by the Azuma-Hoeffding inequality we have that, with probability at least $1 - \delta/(2T^2)$, the following holds

$$\sum_{t \in I} \mathbb{E}_{x \sim \xi} \left[ \bar{f}(x) \right] - \sum_{t \in I} \mathbb{E}_{x \sim \xi} \left[ f_t(x) \right] \leq \sqrt{2|I| \log \left( \frac{2T^2}{\delta} \right)} \leq \sqrt{4 T \log \left( \frac{2T}{\delta} \right)}.$$

By taking a union bound over all possible $T^2$ intervals, we obtain that, for all possible intervals $I$, the equation above holds with probability $1 - \delta/2$.

The Lemma follows by a union bound on the two sets of equations above. $\qquad \square$

These auxiliary technical lemmas are used in proving the following result.

**Lemma 6.2** (Self-bounding lemma). *Let $\eta_{OGD} := \left( 800 \cdot m \cdot \max \left\{ \overline{R^p}_{T, \delta}(\mathcal{X}), E_{T, \delta} \right\} \right)^{-1}$, then if* Alg$_D$ *is* OGD *on the set $\mathcal{D} = \mathbb{R}^m_{\geq 0}$, and the primal algorithm* Alg$_P$ *is 2-scale-free and has a high-probability weakly adaptive regret bound $\overline{R^p}_{T, \delta}(\mathcal{X})$, then with probability at least $1 - \delta$:*

$$\max_{t \in [\![T]\!]} \|\boldsymbol{\lambda}_t\|_1 \leq \frac{13m}{\rho},$$

*where $\rho = \rho_{Adv}$ or $\rho = \rho_{Stoc}$ depending on the setting and $E_{T, \delta} := \sqrt{16 T \log \left( 2T/\delta \right)}$.*

*Proof.* Let $c_1 := 2$ and $c_2 := 13m$ and any learning rate $\eta$ for OGD with $\eta \leq \eta_{OGD}$. By contradiction, suppose there exists a time such that $\|\boldsymbol{\lambda}_t\|_1 \geq c_2/\rho$, and let $t_2 \in [\![T]\!]$ be the smallest $t$ for which this happens. We unify the proof of the adversarial and stochastic setting. In particular, let $\rho = \rho_{\text{Adv}}$ if the losses $(f_t, \boldsymbol{g}_t)$ are adversarial, and let $\rho = \rho_{\text{Stoc}}$ if $(f_t, \boldsymbol{g}_t)$ are

stochastic with mean $(\bar{f}, \bar{g})$. The extra stochasticity coming from the environment in the stochastic setting will be handled through Lemma B.3. In order to streamline the notation, we define $E_{T,\delta} := \sqrt{16T \log{(2T/\delta)}}$.

Then, let $t_1 \in [\![t_2]\!]$ be the largest time smaller than $t_2$ such that $\|\boldsymbol{\lambda}_{t_1}\| \geq \frac{c_1}{\rho}$ and $\|\boldsymbol{\lambda}_{t_1-1}\| \leq \frac{c_1}{\rho}$.

**Step 1.** First, we need to bound $\|\boldsymbol{\lambda}_{t_1}\|_1$ and $\|\boldsymbol{\lambda}_{t_2}\|_1$. To do that, we exploit Lemma B.1. In particular, by telescoping the sum in the lemma, we obtain that:

$$\|\boldsymbol{\lambda}_{t_2}\|_1 - \|\boldsymbol{\lambda}_{t_1}\|_1 \leq \eta m(t_2 - t_1).$$

Moreover, by the definition of $\boldsymbol{\lambda}_{t_1}$ and $\boldsymbol{\lambda}_{t_2}$, we have:

$$\frac{c_1}{\rho} \leq \|\boldsymbol{\lambda}_{t_1}\|_1 \leq \|\boldsymbol{\lambda}_{t_1-1}\|_1 + m\eta \leq \frac{c_1}{\rho} + m\eta$$

and similarly

$$\frac{c_2}{\rho} \leq \|\boldsymbol{\lambda}_{t_2}\|_1 \leq \|\boldsymbol{\lambda}_{t_2-1}\|_1 + m\eta \leq \frac{c_2}{\rho} + m\eta.$$

This, together with the inequality above, yields

$$\frac{c_2 - c_1}{2\eta m\rho} \leq t_2 - t_1. \tag{3}$$

**Step 2.** The range of the primal utilities in the turns $[\![t_1, t_2]\!]$ can now be bounded as:

$$\sup_{x \in \mathcal{X}, t \in [\![t_1, t_2]\!]} |u_t^{\mathbb{P}}(x)| \leq \sup_{x \in \mathcal{X}, t \in [\![t_1, t_2]\!]} \{|f_t(x)| + \|\lambda_t\|_1 \cdot \|\boldsymbol{g}_t(x)\|_\infty\}$$

$$\leq 1 + \frac{c_2}{\rho} + m\eta$$

$$\leq 1 + \frac{12m + 1}{\rho}$$

$$\leq \frac{14m}{\rho} =: L.$$

Now, by the assumption that $\mathtt{Alg}_{\mathbb{P}}$ is weakly adaptive and 2-scale-free, we obtain:

$$R_{[\![t_1, t_2]\!]}^{\mathbb{P}}(\mathcal{X}) \leq L^2 \cdot \overline{R^{\mathbb{P}}}_{T,\delta}(\mathcal{X}),$$

which holds with probability at least $1 - \delta$.

If we apply the primal no-regret condition above for strictly safe strategy $\xi^\circ \in \Delta(\mathcal{X})$ we have

$$\sum_{t \in [\![t_1, t_2]\!]} \mathcal{L}_{f_t, \boldsymbol{g}_t}(x_t, \boldsymbol{\lambda}_t) \geq \mathbb{E}_{x \sim \xi^\circ} \left[ \sum_{t \in [\![t_1, t_2]\!]} \mathcal{L}_{f_t, \boldsymbol{g}_t}(x, \boldsymbol{\lambda}_t) \right] - L^2 \overline{R^{\mathbb{P}}}_{T,\delta}(\mathcal{X}). \tag{4}$$

Moreover, by definition of safe strategy we have that in the adversarial setting $\mathbb{E}_{x \sim \xi^\circ}[g_{t,i}(x)] \leq -\rho_{\mathtt{Adv}}$ for all $i \in [\![m]\!]$ and $t \in [\![t_1, t_2]\!]$, while in the stochastic setting by Lemma B.3 it holds

$$\sum_{t \in [\![t_1, t_2]\!]} \mathbb{E}_{x \sim \xi^\circ}[\langle \boldsymbol{\lambda}_t, \boldsymbol{g}_t(x) \rangle] \leq \sum_{t \in [\![t_1, t_2]\!]} \mathbb{E}_{x \sim \xi^\circ}[\langle \boldsymbol{\lambda}_t, \bar{\boldsymbol{g}}_t(x) \rangle] + ME_{T,\delta}$$

and

$$\mathbb{E}_{x \sim \xi^\circ}[\bar{g}_i(\xi)] \leq -\rho_{\mathtt{Stoc}} \quad \forall i \in [\![m]\!],$$

where we recall that $E_{T,\delta} = \sqrt{16T \log{(2T/\delta)}}$ and $M = \sup_{t \in [\![T]\!]} \|\boldsymbol{\lambda}\|_1$.

Therefore, we can lower bound the first term of the right-hand side of Equation (4) the stochastic setting as:

$$\mathbb{E}_{x \sim \xi^\circ}\left[\sum_{t \in [\![t_1, t_2]\!]} \mathcal{L}_{f_t, \boldsymbol{g}_t}(x, \boldsymbol{\lambda}_t)\right] = \mathbb{E}_{x \sim \xi^\circ}\left[\sum_{t \in [\![t_1, t_2]\!]} f_t(x) - \langle \boldsymbol{\lambda}_t, \boldsymbol{g}_t(x)\rangle\right]$$

$$\geq -\mathbb{E}_{x \sim \xi^\circ}\left[\langle \boldsymbol{\lambda}_t, \boldsymbol{g}_t(x)\rangle\right]$$

$$\geq -\mathbb{E}_{x \sim \xi^\circ}\left[\langle \boldsymbol{\lambda}_t, \bar{\boldsymbol{g}}(x)\rangle\right] - \left(\sup_{t \in [\![T]\!]} \|\boldsymbol{\lambda}\|_1\right) E_{T,\delta}$$

$$\geq \rho_{\text{Stoc}} \sum_{t \in [\![t_1, t_2]\!]} \|\boldsymbol{\lambda}_t\|_1 - \left(\sup_{t \in [\![T]\!]} \|\boldsymbol{\lambda}\|_1\right) E_{T,\delta}$$

$$\geq \rho_{\text{Stoc}} \sum_{t \in [\![t_1, t_2]\!]} \|\boldsymbol{\lambda}_t\|_1 - \left(\frac{c_2}{\rho_{\text{Stoc}}} + m\eta\right) E_{T,\delta}$$

$$\geq c_1(t_2 - t_1) - \left(\frac{c_2}{\rho_{\text{Stoc}}} + m\eta\right) E_{T,\delta}$$

In the adversarial setting we can more easily conclude that $\mathbb{E}_{x \sim \xi^\circ}\left[\sum_{t \in [\![t_1, t_2]\!]} \mathcal{L}_{f_t, \boldsymbol{g}_t}(x, \boldsymbol{\lambda}_t)\right] \geq c_1(t_2 - t_1)$ and thus in both settings it holds that:

$$\mathbb{E}_{x \sim \xi^\circ}\left[\sum_{t \in [\![t_1, t_2]\!]} \mathcal{L}_{f_t, \boldsymbol{g}_t}(x, \boldsymbol{\lambda}_t)\right] \geq c_1(t_2 - t_1) - \left(\frac{c_2}{\rho_{\text{Stoc}}} + m\eta\right) E_{T,\delta}. \tag{5}$$

Combining the two inequalities of Equation (4) and Equation (5), we can conclude that the overall utility of the primal algorithm $\texttt{Alg}_\texttt{P}$ can be lower bounded by:

$$\sum_{t \in [\![t_1, t_2]\!]} u_t^{\texttt{P}}(x_t) \geq c_1(t_2 - t_1) - L^2 \overline{R^{\texttt{P}}}_{T,\delta}(\mathcal{X}) - \left(\frac{c_2}{\rho} + m\eta\right) E_{T,\delta} \tag{6}$$

Now, we need an auxiliary result that we will use to upper bound the left hand side of the previous inequality.

**Claim B.4.** *It holds that:*
$$\sum_{t \in [\![t_1, t_2]\!]} \langle \boldsymbol{\lambda}_t, \boldsymbol{g}_t(x_t)\rangle \geq \frac{m}{2\rho^2\eta}.$$

Then, we upper bound the left-hand side by using Claim B.4:

$$\sum_{t \in [\![t_1, t_2]\!]} u_t^{\texttt{P}}(x_t) = \sum_{t \in [\![t_1, t_2]\!]} \mathcal{L}_{f_t, \boldsymbol{g}_t}(x_t, \boldsymbol{\lambda}_t) = \sum_{t \in [\![t_1, t_2]\!]} [f_t(x_t) - \langle \boldsymbol{\lambda}_t, \boldsymbol{g}_t(x_t)\rangle]$$

$$\leq (t_2 - t_1) - \frac{m}{2\rho^2\eta} \tag{7}$$

Thus, combining Equation (7) and (6)

$$t_2 - t_1 \leq \frac{1}{c_1 - 1}\left(L^2 \overline{R^{\texttt{P}}}_{T,\delta}(\mathcal{X}) - \frac{m}{2\rho^2\eta} + \left(\frac{c_2}{\rho} + m\eta\right) E_{T,\delta}\right).$$

Combining it with Equation (3) one obtains that:

$$\frac{c_2 - c_1}{2\eta m\rho} \leq \frac{1}{c_1 - 1}\left(L^2 \overline{R^{\texttt{P}}}_{T,\delta}(\mathcal{X}) - \frac{m}{2\rho^2\eta} + \left(\frac{c_2}{\rho} + m\eta\right) E_{T,\delta}\right),$$

which gives as a solution $\eta \geq \frac{m^2 - 2\rho + 13m\rho}{392m^3 \overline{R^{\mathbb{P}}}_{T,\delta}(\mathcal{X} + 2m\rho E_{T,\delta}(1+13m))}$. Which is a contradiction since:

$$\eta \leq \eta_{\text{OGD}} := \frac{1}{800 \cdot m \cdot \max\left\{\overline{R^{\mathbb{P}}}_{T,\delta}(\mathcal{X}), E_{T,\delta}\right\}} > \frac{m^2 - 2\rho + 13m\rho}{392m^3 \overline{R^{\mathbb{P}}}_{T,\delta}(\mathcal{X} + 2m\rho E_{T,\delta}(1+13m))}$$

Thus, we can conclude that $\|\boldsymbol{\lambda}_t\|_t \leq c_2/\rho$ for each $t \in [\![T]\!]$. $\hfill\square$

Now, we provide the proof of Claim B.4.

***Proof of Claim B.4.*** We define $\tilde{t}_i$ as the last time in $[\![t_1, t_2]\!]$ in which $\lambda_{\tilde{t}_{i,1}} = 0$, or $\tilde{t}_{1,i} = t_1$ if $\lambda_{t,i} > 0$ for all $t \in [\![t_1, t_2]\!]$. Formally:

$$\tilde{t}_{1,i} = \max\left\{t_1, \sup_{\tau \in [\![t_2]\!]:\lambda_{\tau,i}=0} \tau\right\}.$$

We are now going to analyze separately for all $i \in [\![m]\!]$, the rounds $[\![t_1, \tilde{t}_{1,i}]\!]$ and the rounds $[\![\tilde{t}_{1,i}, t_2]\!]$.

**Phase 1:** First, we analyze the rounds $[\![t_1, \tilde{t}_{1,i}]\!]$. By definition, it can be either that $\lambda_{\tilde{t}_{1,i}} = 0$ or $\tilde{t}_{1,i} = t_1$. In the latter case, $[\![t_1, \tilde{t}_{1,i}]\!] = \emptyset$ and the dual algorithm incurs zero regret. In the former case, we can use Lemma B.2 and write that the regret over the interval with respect to $\lambda_i^* = 0$ is

$$0 \leq \sum_{t \in [\![t_1, \tilde{t}_{1,i}]\!]} \lambda_{t,i} g_{t,i}(x_t) + \frac{\lambda_{t_1}^2}{2\eta} + \frac{1}{2}\eta T \leq \sum_{t \in [\![t_1, \tilde{t}_{1,i}]\!]} \lambda_{t,i} g_{t,i}(x_t) + \frac{\lambda_{t_1}^2}{2\eta} + \frac{1}{2}\eta T. \tag{8}$$

**Phase 2:** Now, we consider the rounds $[\![\tilde{t}_{1,i}, t_2]\!]$. We take $\boldsymbol{\lambda}^*$ defined as follows: $\lambda_i^* = \frac{1}{\rho}$ for all $i \in [\![m]\!]$.

Let $\widetilde{\Delta}_i := \lambda_{t_2,i} - \lambda_{\tilde{t}_{1,i},i}$. Due to the definition of $\tilde{t}_{1,i}$, gradient descent never projects the multiplier relative to constraint $i$, and we can write that

$$\sum_{t \in [\![\tilde{t}_{1,i}, t_2]\!]} g_{t,i}(x_t) = \frac{\widetilde{\Delta}_i}{\eta}$$

and, therefore,

$$\sum_{t \in [\![\tilde{t}_{1,i}, t_2]\!]} \lambda_i^* g_{t,i}(x_t) = \frac{\widetilde{\Delta}_i}{\rho\eta}. \tag{9}$$

Now we can use Lemma B.2 to find that:

$$\sum_{t \in [\![\tilde{t}_{1,i}, t_2]\!]} \lambda_i^* g_{t,i}(x_t) \leq \sum_{t \in [\![\tilde{t}_1, t_2]\!]} \lambda_{t,i} g_{t,i}(x_t) + \frac{(\lambda_i^* - \lambda_{\tilde{t}_{1,i},i})^2}{2\eta} + \frac{1}{2}\eta T.$$

Combining it with Equation (9) yields the following

$$\sum_{t \in [\![\tilde{t}_{1,i}, t_2]\!]} \lambda_{t,i} g_{t,i}(x_t) \geq \frac{\widetilde{\Delta}_i}{\rho\eta} - \frac{(\lambda_i^* - \lambda_{\tilde{t}_{1,i},i})^2}{2\eta} - \frac{1}{2}\eta T. \tag{10}$$

Combining Equation (10) and Equation (8) we obtain:

$$\sum_{t \in [\![t_1, t_2]\!]} \lambda_{t,i} g_{t,i}(x_t) \geq \frac{\widetilde{\Delta}_i}{\rho\eta} - \frac{(\lambda_i^* - \lambda_{\tilde{t}_{1,i},i})^2}{2\eta} - \frac{\lambda_{t_1}^2}{2\eta} - \eta T$$

$$\geq \frac{\widetilde{\Delta}_i}{\rho\eta} - \frac{(\lambda_i^*)^2 + \lambda_{\tilde{t}_{1,i},i}^2}{2\eta} - \frac{\lambda_{t_1}^2}{2\eta} - \eta T.$$

Now, by summing over all $i \in [\![m]\!]$, and by letting $\boldsymbol{\lambda}_{\tilde{t}_1}$ be the vector that has $\lambda_{\tilde{t}_1,i}$ as its $i$-th component, we get:

$$\sum_{t \in [\![t_1,t_2]\!]} \langle \boldsymbol{\lambda}_t, \boldsymbol{g}_t(x_t) \rangle \geq \frac{\|\boldsymbol{\lambda}_{t_2}\|_1 - \|\boldsymbol{\lambda}_{\tilde{t}_1}\|_1}{\rho\eta} - \frac{1}{2\eta}\left(\|\boldsymbol{\lambda}^*\|_2^2 + \|\boldsymbol{\lambda}_{\tilde{t}_1}\|_2^2 + \|\boldsymbol{\lambda}_{t_1}\|_2^2\right) - \frac{1}{\eta} \quad (\text{as } \eta \leq 1/\sqrt{T})$$

$$\geq \frac{c_2}{\rho^2\eta} - \frac{1}{\rho\eta}\|\boldsymbol{\lambda}_{t_1}\|_1 - \frac{1}{2\eta}\left(\|\boldsymbol{\lambda}^*\|_2^2 + 2\|\boldsymbol{\lambda}_{t_1}\|_2^2\right) - \frac{1}{\eta} \quad (\|\boldsymbol{\lambda}\|_1 \geq c_2/\rho \text{ and } \|\boldsymbol{\lambda}_{\tilde{t}_1}\|_1 \leq \|\boldsymbol{\lambda}_{t_1}\|_1)$$

$$\geq \frac{c_2}{\rho^2\eta} - \frac{1}{\rho\eta}\left(\frac{c_1}{\rho} + m\eta\right) - \frac{1}{2\eta}\left(\frac{m}{\rho^2} + 2\left(\frac{c_1}{\rho} + m\eta\right)^2\right) - \frac{1}{\eta}$$

$$\geq \frac{c_2}{\rho^2\eta} - \frac{c_1+1}{\rho^2\eta} - \frac{m}{2\rho^2\eta} - \frac{2(c_1+1)^2}{2\rho^2\eta} - \frac{1}{\eta} \quad (\eta \leq 1/\rho m)$$

$$\geq \frac{2c_2 - 24 - m}{2\rho^2\eta}$$

$$\geq \frac{m}{2\rho^2\eta}$$

where the last two inequalities hold due to the choice of parameters in the proof of Claim B.4, that is $c_1 = 2$ and $c_2 = 13m$. This concludes the proof. $\qquad\square$

## C. Omitted Proofs from Section 7

**Theorem 7.1.** *Let* `Alg`$_D$ *be* `OGD` *with learning rate $\eta$ as in Lemma 6.2, and let* `Alg`$_P$ *any 2-scale-free algorithm with no-adaptive regret. Then, with probability at least $1 - \delta$, it holds that*

$$V_T = \widetilde{O}\left(\frac{m^2}{\rho}\overline{R^P}_{T,\delta}(\mathcal{X})\right),$$

*where $\rho = \rho_{Adv}$ in the adversarial setting and $\rho = \rho_{Stoc}$ in the stochastic.*

*Proof.* The update of `OGD` for each component $i \in [\![m]\!]$ is $\lambda_{t+1,i} := [\lambda_{t,i} + \eta \boldsymbol{g}_{t,i}(x_t)]^+$. Thus:

$$\lambda_{t+1,i} \geq \lambda_{t,i} + \eta_{\text{OGD}} g_{t,i}(x_t),$$

and by induction:

$$\lambda_{t+1,i} \geq \lambda_{0,i} + \eta_{\text{OGD}} \sum_{\tau=1}^{t} g_{\tau,i}(x_\tau).$$

By rearranging and recalling that $\lambda_{0,i} = 0$ we obtain:

$$\sum_{t \in [\![T]\!]} g_{t,i}(x_t) \leq \frac{1}{\eta_{\text{OGD}}}\lambda_{T+1,i} \leq \frac{1}{\eta}\|\boldsymbol{\lambda}_{T+1}\|_1$$

Moreover, by Lemma 6.2 we can bound $\|\boldsymbol{\lambda}_T\|_1 \leq \frac{13m}{\rho}$ which holds with probability at least $1 - \delta$. Thus, with probability at least $1 - \delta$, it holds:

$$V_T := \max_{i \in [\![m]\!]} V_i(T) \leq \frac{13m}{\eta_{\text{OGD}}\rho}.$$

The proof is concluded by observing that $\eta_{\text{OGD}} = \tilde{O}\left((m\overline{R^P}_{T,\delta}(\mathcal{X}))^{-1}\right)$. $\qquad\square$

**Theorem 7.2.** *If* `Alg`$_D$ *is* `OGD` *with learning rate $\eta_{OGD}$ and domain $\mathcal{D} := \mathbb{R}_{\geq 0}^m$, and* `Alg`$_P$ *is 2-scale-free, then, in the adversarial setting, with high probability:*

$$Rew \geq \frac{\rho_{Adv}}{1 + \rho_{Adv}}Opt_{Adv} - \widetilde{O}\left(\left(\frac{m}{\rho_{Adv}}\right)^2 \overline{R^P}_{T,\delta}(\mathcal{X})\right).$$

*Proof.* Define $x^* \in \mathcal{X}$ such that:

$$\sum_{t \in [\![T]\!]} f_t(x^*) = \mathrm{Opt}_{\mathrm{Adv}}$$

Now, consider a randomized strategy $\xi$ that randomized with probability $\alpha$ between $x^*$ and $\xi^\circ$, where $\xi^\circ$ is any strategy for which $\mathbb{E}_{x \sim \xi^\circ}[g_{t,i}(x_t)] \leq -\rho_{\mathrm{Adv}}$. This strategy exists by assumption. Formally, for any $x \in \mathcal{X}$ the randomized strategy $\xi$ assigns probability to $x$:

$$\xi(x) = \alpha \delta_{x^*}(x) + (1 - \alpha)\xi^\circ(x).$$

Then, we compute the component of the primal utility of $\xi$ due to a constraint $i \in [\![m]\!]$ as follows:

$$\mathbb{E}_{x \sim \xi}\left[\sum_{t \in [\![T]\!]} \lambda_{t,i} g_{t,i}(x)\right] = \alpha \sum_{t \in [\![T]\!]} \lambda_{t,i} g_{t,i}(x^*) + (1 - \alpha)\mathbb{E}_{x \sim \xi^\circ}\left[\sum_{t \in [\![T]\!]} \lambda_{t,i} g_{t,i}(x)\right]$$

$$\leq \alpha \sum_{t \in [\![T]\!]} \lambda_{t,i} - (1 - \alpha)\rho_{\mathrm{Adv}} \sum_{t \in [\![T]\!]} \lambda_{t,i}$$

$$\leq (\alpha - (1 - \alpha)\rho_{\mathrm{Adv}}) \sum_{t \in [\![T]\!]} \lambda_{t,i}.$$

Thus, setting $\alpha = \frac{\rho_{\mathrm{Adv}}}{1 + \rho_{\mathrm{Adv}}}$ we have that $\mathbb{E}_{x \sim \xi}\left[\sum_{t \in [\![T]\!]} \lambda_{t,i} g_{t,i}(x)\right] \leq 0$, and $\sum_{t \in [\![T]\!]} \langle \boldsymbol{\lambda}_t, \boldsymbol{g}_t(x_t) \rangle \leq 0$.

We now compute the reward of $\xi$ for $\alpha = \frac{\rho_{\mathrm{Adv}}}{1 + \rho_{\mathrm{Adv}}}$:

$$\mathbb{E}_{x \sim \xi}\left[\sum_{t \in [\![T]\!]} f_t(x)\right] = \alpha \sum_{t \in [\![T]\!]} f_t(x^*) + (1 - \alpha)\mathbb{E}_{x \sim \xi^\circ}\left[\sum_{t \in [\![T]\!]} f_t(x)\right]$$

$$\geq \frac{\rho_{\mathrm{Adv}}}{1 + \rho_{\mathrm{Adv}}} \mathrm{Opt}_{\mathrm{Adv}}$$

Now, we consider the regret of $\mathtt{Alg_P}$ with respect to $\xi$ and we find that:

$$\sum_{t \in [\![T]\!]} \mathcal{L}_{f_t, \boldsymbol{g}_t}(x_t, \boldsymbol{\lambda}_t) \geq \mathbb{E}_{x \sim \xi}\left[\sum_{t \in [\![T]\!]} \mathcal{L}_{f_t, \boldsymbol{g}_t}(x, \boldsymbol{\lambda}_t)\right] - L^2 \cdot \overline{R^{\mathrm{P}}}_{T,\delta}(\mathcal{X}).$$

where $L$ is the maximum module of the payoffs of the primal regret minimizer, *i.e.,* $L :- \sup_{t \in [\![T]\!], x \in \mathcal{X}} |u_t^{\mathrm{P}}(x)|$.

Exploiting the definition of $\mathcal{L}_{f_t, \boldsymbol{g}_t}(\cdot, \cdot)$ in the inequality above we obtain that:

$$\sum_{t \in [\![T]\!]} f_t(x_t) - \langle \boldsymbol{\lambda}_t, \boldsymbol{g}_t(x_t) \rangle \geq \mathbb{E}_{x \sim \xi}\left[\sum_{t \in [\![T]\!]} f_t(x) - \langle \boldsymbol{\lambda}_t, \boldsymbol{g}_t(x) \rangle\right] - L^2 \cdot \overline{R^{\mathrm{P}}}_{T,\delta}(\mathcal{X})$$

$$\geq \mathbb{E}_{x \sim \xi}\left[\sum_{t \in [\![T]\!]} f_t(x)\right] - L^2 \cdot \overline{R^{\mathrm{P}}}_{T,\delta}(\mathcal{X})$$

$$\geq \frac{\rho_{\mathrm{Adv}}}{1 + \rho_{\mathrm{Adv}}} \mathrm{Opt}_{\mathrm{Adv}} - L^2 \cdot \overline{R^{\mathrm{P}}}_{T,\delta}(\mathcal{X}) \tag{11}$$

Then, we lower bound the term $\sum_{t \in [\![T]\!]} \langle \boldsymbol{\lambda}_t, \boldsymbol{g}_t(x_t) \rangle$ by using the dual regret of $\mathtt{Alg_D}$ with respect to $\boldsymbol{\lambda}^* = \boldsymbol{0}$. Indeed,

$$\sum_{t \in [\![T]\!]} \langle \boldsymbol{\lambda}^* - \boldsymbol{\lambda}_t, \boldsymbol{g}_t(x_t) \rangle \leq \overline{R^{\mathrm{D}}}_{T,\delta}(\{\boldsymbol{\lambda}^*\})$$

implies that

$$\sum_{t \in [\![T]\!]} \langle \boldsymbol{\lambda}_t, \boldsymbol{g}_t(x_t) \rangle \geq -\overline{R^{\mathrm{D}}}_{T,\delta}(\{\boldsymbol{\lambda}^*\}).$$

Combining it with Equation (11) gives:

$$\sum_{t \in [\![T]\!]} f_t(x_t) \geq \frac{\rho_{\mathrm{Adv}}}{1 + \rho_{\mathrm{Adv}}} \mathrm{Opt}_{\mathrm{Adv}} - L^2 \cdot \overline{R^{\mathrm{P}}}_{T,\delta}(\mathcal{X}) - \overline{R^{\mathrm{D}}}_{T,\delta}(\{\boldsymbol{\lambda}^*\}).$$

Now, we use Lemma 6.2 which bounds $L \leq 2\frac{13m}{\rho_{\mathrm{Adv}}}$ and Lemma B.1 which we can use to bound $\overline{R^{\mathrm{D}}}_{T,\delta}(\{\boldsymbol{\lambda}^*\})$.

In particular, $\overline{R^{\mathrm{D}}}_{T,\delta}(\{\boldsymbol{\lambda}^*\})$ can be bounded with:

$$\overline{R^{\mathrm{D}}}_{T,\delta}(\{\boldsymbol{\lambda}^*\}) \leq \frac{1}{2}\eta_{\mathrm{OGD}}mT,$$

and thus:

$$\mathrm{Rew} := \sum_{t \in [\![T]\!]} f_t(x_t) \geq \frac{\rho_{\mathrm{Adv}}}{1 + \rho_{\mathrm{Adv}}} \mathrm{Opt}_{\mathrm{Adv}} - 676 \left( \frac{m}{\rho_{\mathrm{Adv}}} \right)^2 \overline{R^{\mathrm{P}}}_{T,\delta}(\mathcal{X}) - \eta_{\mathrm{OGD}}mT.$$

The proof is concluded by noting that $\eta_{\mathrm{OGD}} = \tilde{O}\left((m\overline{R^{\mathrm{P}}}_{T,\delta}(\mathcal{X}))^{-1}\right)$. $\qquad\square$

**Theorem 7.3.** *If $\mathtt{Alg}_{\mathrm{D}}$ is OGD with learning rate $\eta_{\mathrm{OGD}}$ and domain $\mathcal{D} := \mathbb{R}_{\geq 0}^m$, and $\mathtt{Alg}_{\mathrm{P}}$ is 2-scale-free, then in the stochastic setting, in high probability:*

$$\mathrm{Rew} \geq \mathrm{Opt}_{\mathit{Stoc}} - \tilde{O}\left( \left( \frac{m}{\rho_{\mathit{Stoc}}} \right)^2 \overline{R^{\mathrm{P}}}_{T,\delta}(\mathcal{X}) \right).$$

*Proof.* By Lemma 6.2 we have that with probability at least $1 - \delta$ we have that $\sup_{t \in [\![T]\!]} \|\boldsymbol{\lambda}_t\|_1 \leq \frac{13m}{\rho_{\mathrm{Stoc}}}$ and in the same way $\sup_{t \in [\![T]\!], x \in \mathcal{X}} \|u_t^{\mathrm{P}}(x)\|_1 \leq 2\frac{13m}{\rho_{\mathrm{Stoc}}}$.

Define $\xi$ as the best strategy that satisfies the constraints, *i.e.*, $\mathrm{Opt}_{\mathrm{Stoc}} := T\, \mathbb{E}_{x \sim \xi}\left[\bar{f}(x)\right]$ and $\mathbb{E}_{x \sim \xi}[\bar{g}_i(x)] \leq 0$. The no-regret property of $\mathtt{Alg}_{\mathrm{P}}$ with respect to $\xi$ gives that with probability $1 - \delta$ it holds:

$$\sum_{t \in [\![T]\!]} \left[ f_t(x_t) - \langle \boldsymbol{\lambda}_t, \boldsymbol{g}_t(x_t) \rangle \right]$$

$$\geq \mathbb{E}_{x \sim \xi} \left[ \sum_{t \in [\![T]\!]} \left[ f_t(x) - \langle \boldsymbol{\lambda}_t, \boldsymbol{g}_t(x) \rangle \right] \right] - \left( 2\frac{13m}{\rho_{\mathrm{Stoc}}} \right)^2 \overline{R^{\mathrm{P}}}_{T,\delta}(\mathcal{X})$$

$$\geq \mathbb{E}_{x \sim \xi} \left[ \sum_{t \in [\![T]\!]} \left[ \bar{f}(x) - \langle \boldsymbol{\lambda}_t, \bar{\boldsymbol{g}}(x) \rangle \right] \right] - 676 \left( \frac{m}{\rho_{\mathrm{Stoc}}} \right)^2 \overline{R^{\mathrm{P}}}_{T,\delta}(\mathcal{X}) - 2 \left( \frac{13m}{\rho_{\mathrm{Stoc}}} \right) E_{T,\delta}$$

$$= T\, \mathrm{Opt}_{\mathrm{Stoc}} - 676 \left( \frac{m}{\rho_{\mathrm{Stoc}}} \right)^2 \overline{R^{\mathrm{P}}}_{T,\delta}(\mathcal{X}) - \frac{26m}{\rho_{\mathrm{Stoc}}} E_{T,\delta},$$

where the second inequality follows from Lemma B.3 with $M := \frac{13m}{\rho_{\mathrm{Stoc}}}$.

Moreover, the no-regret property of the dual regret minimizer $\mathtt{Alg}_{\mathrm{D}}$, with respect to $\boldsymbol{\lambda}^* = \boldsymbol{0}$, gives that:

$$\sum_{t \in [\![T]\!]} \langle \boldsymbol{\lambda}^* - \boldsymbol{\lambda}_t, \boldsymbol{g}_t(x_t) \rangle \leq \frac{1}{2}\eta_{\mathrm{OGD}}mT.$$

Finally, we can combine everything from which follows that:

$$\mathrm{Rew} \geq \mathrm{Opt}_{\mathrm{Stoc}} - 676 \left( \frac{m}{\rho_{\mathrm{Stoc}}} \right)^2 \overline{R^{\mathrm{P}}}_{T,\delta}(\mathcal{X}) - \frac{26m}{\rho_{\mathrm{Stoc}}} E_{T,\delta} - \frac{1}{2}\eta_{\mathrm{OGD}}mT.$$

The proof is concluded by observing that $\eta_{\mathrm{OGD}} = \tilde{O}\left((m\overline{R^{\mathrm{P}}}_{T,\delta}(\mathcal{X}))^{-1}\right)$ and $E_{T,\delta} = \tilde{O}(\sqrt{T})$ $\qquad\square$

# D. Proofs omitted from Section 8

**Lemma 8.3.** *The error of $\mathcal{O}_{\mathcal{L}}$ can be bounded as*

$$Err(\mathcal{O}_{\mathcal{L}}) \leq 2Err(\mathcal{O}_f) + 2 \left( \sup_{t \in [\![T]\!]} \|\boldsymbol{\lambda}_t\|_1 \right)^2 \sum_{i \in [\![m]\!]} Err(\mathcal{O}_i).$$

*Proof.* Consider the following inequalities:

$$
\begin{aligned}
\texttt{Err}(\mathcal{O}_{\mathcal{L}}) &:= \sum_{t \in [\![T]\!]} \left( \hat{\mathcal{L}}_t(z_t, a_t) - \bar{\mathcal{L}}(z_t, a_t) \right)^2 \\
&\leq 2 \sum_{t \in [\![T]\!]} \left( \hat{f}_t(z_t, a_t) - \bar{f}(z_t, a_t) \right)^2 + 2 \sum_{t \in [\![T]\!]} \left( \langle \boldsymbol{\lambda}_t, \hat{\boldsymbol{g}}_t(z_t, a_t) \rangle - \langle \boldsymbol{\lambda}_t \bar{\boldsymbol{g}}(z_t, a_t) \rangle \right)^2 \\
&\qquad\qquad\qquad\qquad\qquad\qquad\qquad \text{(By AM-GM inequality: } 2ab \leq a^2 + b^2 \text{ for } a, b \geq 0.) \\
&= 2 \cdot \texttt{Err}(\mathcal{O}_f) + 2 \sum_{t \in [\![T]\!]} \left( \langle \boldsymbol{\lambda}_t, \hat{\boldsymbol{g}}_t(z_t, a_t) - \bar{\boldsymbol{g}}(z_t, a_t) \rangle \right)^2 \\
&\leq 2 \cdot \texttt{Err}(\mathcal{O}_f) + 2 \sum_{t \in [\![T]\!]} \|\boldsymbol{\lambda}_t\|_1^2 \cdot \|\hat{\boldsymbol{g}}_t(z_t, a_t) - \bar{\boldsymbol{g}}(z_t, a_t)\|_\infty^2 \qquad (\langle a, b \rangle \leq \|a\|_1 \cdot \|b\|_\infty) \\
&\leq 2 \cdot \texttt{Err}(\mathcal{O}_f) + 2 \left( \sup_{t \in [\![T]\!]} \|\boldsymbol{\lambda}_t\|_1 \right)^2 \cdot \sum_{t \in [\![T]\!]} \|\hat{\boldsymbol{g}}_t(z_t, a_t) - \bar{\boldsymbol{g}}(z_t, a_t)\|_\infty^2 \\
&\leq 2 \cdot \texttt{Err}(\mathcal{O}_f) + 2 \left( \sup_{t \in [\![T]\!]} \|\boldsymbol{\lambda}_t\|_1 \right)^2 \cdot \sum_{t \in [\![T]\!]} \sum_{i \in [\![m]\!]} (\hat{g}_{t,i}(z_t, a_t) - \bar{g}_i(z_t, a_t))^2 \\
&= 2 \cdot \texttt{Err}(\mathcal{O}_f) + 2 \left( \sup_{t \in [\![T]\!]} \|\boldsymbol{\lambda}_t\|_1 \right)^2 \cdot \sum_{i \in [\![m]\!]} \texttt{Err}(\mathcal{O}_i)
\end{aligned}
$$

which concludes the proof. $\qquad\square$

**Lemma 8.4.** *Assume that $\max\{Err(\mathcal{O}_f), Err(\mathcal{O}_i)\} \leq \overline{Err}$. Then, we have that Algorithm 3 with $\eta_{\mathbb{P}} := \sqrt{KT}$ guarantees that*

$$\sup_{I = [\![t_1, t_2]\!]} R_I^{\mathbb{P}}(\Pi) = \tilde{O}\left( m \cdot \overline{Err} \cdot L^2 \cdot \sqrt{KT} \right)$$

*with high probability, where $L := \sup_{t \in [\![T]\!], \pi \in \Pi} |u_t^{\mathbb{P}}(\pi)|$.*

*Proof.* Consider any interval $I = [\![t_1, t_2]\!] \subseteq [\![T]\!]$. Since the prediction error at each time $t$ is positive, one trivially has that:

$$\sum_{t \in [\![t_1, t_2]\!]} \left( \hat{\mathcal{L}}_t(z_t, a_t) - \bar{\mathcal{L}}(z_t, a_t) \right)^2 \leq \texttt{Err}(\mathcal{O}_{\mathcal{L}}).$$

Then, applying Lemma 8.3 we have that:

$$\sum_{t \in [\![t_1, t_2]\!]} \left( \hat{\mathcal{L}}_t(z_t, a_t) - \bar{\mathcal{L}}(z_t, a_t) \right)^2 \leq 2\texttt{Err}(\mathcal{O}_f) + 2 \sup_{t \in [\![T]\!]} \|\boldsymbol{\lambda}_t\|_1^2 \sum_{i \in [\![m]\!]} \texttt{Err}(\mathcal{O}_i).$$

Moreover, by the assumption on the errors of the oracles it holds that:

$$\sum_{t \in [\![t_1, t_2]\!]} \left( \hat{\mathcal{L}}_t(z_t, a_t) - \bar{\mathcal{L}}(z_t, a_t) \right)^2 \leq 2m(1 + \sup_{t \in [\![T]\!]} \|\boldsymbol{\lambda}_t\|_1^2)\overline{\texttt{Err}}. \tag{12}$$

Note that we could pretend that the algorithm starts at any time $t_1 \in [\![T]\!]$, and the same analysis of Theorem 1 by Foster & Rakhlin (2020) would hold, as their algorithm behavior does not depend on its past behavior. Hence, the following holds:

$$
\begin{aligned}
R^{\mathbb{P}}_{[\![t_1,t_2]\!]}(\Pi) &:= \sup_{\pi \in \Pi} \sum_{t \in [\![t_1,t_2]\!]} [u^{\mathbb{P}}_t(\pi) - u^{\mathbb{P}}_t(\pi_t)] \\
&:= \sup_{\pi \in \Pi} \sum_{t \in [\![t_1,t_2]\!]} [\mathcal{L}_t(\pi(z_t)) - \mathcal{L}_t(\pi_t(z_t))] \\
&= \sup_{\pi \in \Pi} \sum_{t \in [\![t_1,t_2]\!]} [\mathcal{L}_t(\pi(z_t)) - \mathcal{L}_t(a_t)] \\
&\leq \frac{\eta_{\mathbb{P}}}{2} \mathrm{Err}(\mathcal{O}_\mathcal{L}) + 4\eta_{\mathbb{P}} \log\left(\frac{2T^2}{\delta}\right) + 2K\frac{T}{\eta_{\mathbb{P}}} + \sqrt{2T \log\left(\frac{2T^2}{\delta}\right)}
\end{aligned}
$$

which holds with probability $1 - \delta/(T^2)$.

Thus, by an union bound, and combining it with Equation (12) we obtain that:

$$
R^{\mathbb{P}}_{[\![t_1,t_2]\!]}(\Pi) \leq \eta_{\mathbb{P}} m(1 + \sup_{t \in [\![T]\!]} \|\boldsymbol{\lambda}_t\|_1^2)\overline{\mathrm{Err}} + 4\eta_{\mathbb{P}} \log\left(\frac{2T^2}{\delta}\right) + 2K\frac{T}{\eta_{\mathbb{P}}} + \sqrt{2T \log\left(\frac{2T^2}{\delta}\right)},
$$

which holds with probability $1 - \delta/T^2$. Finally, by tuning $\eta_{\mathbb{P}} = \sqrt{KT}$ and applying an union bound on all the $T^2$ possible intervals $[\![t_1,t_2]\!]$, we obtain that with probability $1 - \delta$ it holds that:

$$
\sup_{I=[\![t_1,t_2]\!]} R^{\mathbb{P}}_{[\![t_1,t_2]\!]}(\Pi) \leq 504 \cdot m \, \overline{\mathrm{Err}} \, L^2 \log(T^2/\delta)\sqrt{KT}.
$$

$\square$

