# OpenReview forum: "No-Regret is not enough! Bandits with General Constraints through Adaptive Regret Minimization"
_ICML.cc/2025/Conference — ICML 2025 poster_

### Official Review · Reviewer_Ldsh · 2025-03-09

**Overall Recommendation:** 3

**Summary:**

The authors study the BwK setting where a learner is tasked with repeatedly performing actions and gain high cumulative reward while also satisfying multiple general long-term constraints. Specifically, they consider a best-of-both worlds objective in which a given algorithm has to perform optimally whether or not the environment is stochastic or adversarial, and show that if a primal-dual scheme is applied with weakly adaptive regret minimization algorithms, such best-of-both-worlds guarantees are achievable without prior knowledge of the Slater parameter $\rho$ characterizing the problem instance, which is the main contribution of this work over previous works. They establish the fact that OGD with a specific choice of learning rate is indeed such a weakly adaptive regret minimizer which gives an explicit algorithm for the problem. The authors also provide explicit scenarios where their results can be applied, specifically contextual bandits with constraints.

## update after rebuttal:
After reading the other reviews and the authors' comments, my assessment of the paper remains as is.

**Claims And Evidence:**

The claims made in the submission are supported by rigorous proofs provided in the supplementary material.

**Essential References Not Discussed:**

I am not familiar with essential references that were not discussed in this work.

**Experimental Designs Or Analyses:**

N/A

**Methods And Evaluation Criteria:**

N/A

**Other Comments Or Suggestions:**

As mentioned earlier, the paper contains quite a few typos, some of which are listed in the following:

* In Algorithm 1 and Algorithm 2, I believe $\mathbf{c}_t$ should be replaced with $\mathbf{g}_t$.

* In the statement of Theorem 4.1, I believe the last part should have $<$ instead of $>$, as the current statement does not make sense for a lower bound.

* Lines 224-225 - "... it is not required adaptive regret minimization" - wording is confusing here.

**Other Strengths And Weaknesses:**

Strengths:

* The authors present what seems to be the first optimal best-of-both-worlds guarantees for BwK with general constraints with algorithm that doesn't require knowledge of the problem's Slater parameter.

* The contributions are presented clearly.

* The observation that regret minimizers for primal and dual algorithms does not suffice for the adversarial setting seems interesting.

* The upper bounds apply generally in the sense that they are black-box upper bounds which only require the input primal and dual algorithms to be weakly adaptive regret minimizers with the primal algorithm being scale-free, thus allowing for a wide range of algorithms.

Weaknesses:

* The paper contains numerous typos and confusing wording in some places.

* I would have appreciated some analysis sketch in the main text, to give some intuition and also highlight any technical novelties in the analysis.

**Questions For Authors:**

My main question to the authors concerns the technical novelties of this work when compared to previous works. Specifically:

* Is the observation that adaptive regret minimizers are necessary (rather than regret minimizers) a novel observation (referring to the construction in Example 5.2)?

* Is the fact that the benchmark in the adversarial setting isn't required to satisfy the constraints novel for this work? Or is such a benchmark used in previous works of adversarial BwK?

* I would appreciate it if the authors could briefly summarize the technical challenges and novelties of their analysis. Is it mostly the self-bounding lemma? If so, what are the challenges in proving it?

**Relation To Broader Scientific Literature:**

The authors provide adequate references to related previous works in the context of BwK, with a particular focus on recent works by Castiglioni et al. which is the most relevant to this work in studying best-of-both worlds objectives for BwK. The authors make clear how in those previous works, either the Slater parameter $\rho$ has to be known or the constraints have a specific structure, thus emphasizing the generality of their contribution. The authors also sufficiently cite papers relevant to the LagrangeBwK primal-dual framework which they heavily use in this work.

**Theoretical Claims:**

I did not check the correctness of the proofs presented in the submission, however, given my relative familiarity with the research topic, the claims seem sound and I didn't find any soundness issues.

---

> ### Author Rebuttal · Authors · 2025-04-01
>
> We thank the reviewer for the positive feedback about our paper.
>
> * The weak adaptivity property was used in a simplified setting in the very recent paper by Castiglioni et al. (2024). Here, the authors only study the case in which they have a single budget constraint and a single consumption constraint (ROI).
> However, in our paper, we have to deal with an arbitrary number of constraints. In particular, proving lemma 6.1 in this setting is far more challenging. Indeed, the multiple constraints “move” independently but affect the constraints “jointly”. This originates difficult technical challenges that we highlight in more detail in the answer to your third question.
> * Some prior works, such as Balseiro et al. (2020), could, in principle, accommodate stronger benchmarks through a refinement of their analysis. However, this was not explicitly pointed out by the authors, likely due to their adherence to the conventions of the online allocation literature and its standard benchmarks. Some very recent works also highlighted this stronger benchmark (see, e.g., Bernasconi, Martino, et al. (2024)).
> * The proof relies on the key observation that Lagrangian multipliers jointly affect the primal utility but evolve independently. Thus, we need to reason about the joint behavior of the Lagrangian multipliers and look at the L1 norm. In particular, our proof proceeds by contradiction:  if the Lagrangian multipliers exceed a certain threshold, then they must have remained "large" for an extended period of time (Equation 3 in the appendix). Now we can exploit the scale freeness of the primal regret minimizer and its regret property with respect to the action that satisfies the constraints (Equation 4), to claim that the primal utility is large in such interval (Equation 6). However, this is in contradiction with the fact that the dual utility is also large (because of the growth of the Lagrangian multipliers, see Claim B.4). The proof of Claim B.4 is particularly involved, as it needs to analyze the separate behavior of the Lagrangian multipliers relative to different constraints. Indeed, it is possible for the  $\ell_1$ norm of the multipliers to increase without having all individual components growing. This makes it nontrivial to conclude that the dual utility must have increased as well.
>
> **References:**
>
> - Castiglioni, Matteo, et al. "Online learning under budget and ROI constraints via weak adaptivity." ICML 2024
> - Balseiro, Santiago, Haihao Lu, and Vahab Mirrokni. "Dual mirror descent for online allocation problems." ICML 2020
> - Bernasconi, Martino, et al. "Beyond Primal-Dual Methods in Bandits with Stochastic and Adversarial Constraints." NeurIPS 2024

---

### Official Review · Reviewer_w2eb · 2025-03-13

**Overall Recommendation:** 3

**Summary:**

The paper addresses the problem of bandits with general constraints, extending beyond the traditional bandits with knapsacks (BwK) framework. The authors generalize the setting where the learner does not know the Slater's parameter $\rho$ and give an algorithm following the primal-dual framework.
Previous works (Balseiro et al., 2022; Castiglioni et al., 2022a) need to know the exact value of $\rho$ to find the boundedness of dual multipliers.
In this paper, one key contribution is the "self-bounding" lemma for bounding dual variables if both the primal and dual algorithms are weakly adaptive.
The authors provide best-of-both-worlds guarantees (sublinear regret and constraint violations) and applications to contextual bandits with linear constraints (CBwLC). Theoretical results show competitive ratios of $\rho/(1+\rho)$ in adversarial settings and near-optimal regret in stochastic settings.

**Claims And Evidence:**

The claims made in this paper are clear and supported by proof in the appendix. However, there are some typos in the proof, and I listed them in the Theoretical Claims. If the authors can answer them, I will be convinced that the evidence is clear.

**Essential References Not Discussed:**

The key contribution of this paper focuses on developing a near-optimal algorithm for the bandit problem with general constraints. I would like to know how this work is related to Bandit with budgets [Ding et al., 2013] published in AAAI 2013.

**Ethical Review Concerns:**

No ethical concern.

**Experimental Designs Or Analyses:**

No experimental designs are founded in this paper.

**Methods And Evaluation Criteria:**

The authors use the Regret or Reward gap to the optimal strategy, which is standard in the Bandit area.

**Other Comments Or Suggestions:**

This paper is well-written, and it is smooth to go through from the beginning to the end.

**Other Strengths And Weaknesses:**

This work removes the strong assumption that Slate's parameter is known to the learned used by prior works. This can be considered as a more realistic setting.

**Questions For Authors:**

Please look at the Theoretical claims.

**Relation To Broader Scientific Literature:**

The assumption used in this paper is weak and can be applied to many specific cases, such as multi-armed bandit (MAB) and contextual bandit (CB). The weakly adaptive regret bound for primal and dual problems is reasonable since many algorithms satisfy this property, like *EXP3-SIX* for MAB and *Vovk forecaster* for the finite function class. This paper can be used as a fundamental work to develop more realistic algorithms.

**Theoretical Claims:**

I have some questions on the Theoretical proof and I list them below:

- line 660, missing reference in Lemma B.2
- line 711, constant used in $c_2$ is 12, which does not match with 13 used in the lemma statement?
- line 725 - 727, in the proof of Self-bounding lemma (Lemma 6.1), one step shown in the mentioned line is upper bounding $ \| \lambda_{t_1 - 1} \|_1 + m \eta \leq c_1/\rho + m \eta$ because the inequality $ \| \lambda_{t_1 - 1} \|_1 \leq c_1/ \rho $. I am not aware of any properties of $\|\lambda_{t}\|_1$ and if it is not monotonically increasing with $t$,  $\| \lambda_{t_1 - 1}\|_1$ can be larger than $c_1/\rho$ since the definition of $t_1$ is the largest time between $0$ and $t_2$ for which $\|\lambda_{t_1}\|_1 \in [c_1/\rho,  c_2/\rho]$ and we cannot assume when $t \in [0, t_1]$, $\|\lambda_t\|$ is always smaller than $c_1/\rho$. Correct me if I am wrong.

---

> ### Author Rebuttal · Authors · 2025-04-01
>
> **On the questions about the Theoretical proofs:**
>
> Thanks for taking the time to carefully read our proofs. We really appreciate the effort.
>
> * Thanks. We meant to cite the following works:
>     - At line 660: Hazan, Elad. "Introduction to online convex optimization." Foundations and Trends® in Optimization (2016).
>     - At line 1100: Foster, Dylan, and Alexander Rakhlin. "Beyond ucb: Optimal and efficient contextual bandits with regression oracles." International conference on machine learning. PMLR, 2020.
>
> * Yes, you are right; $c_2$ should be 13m.
> * The confusion likely stems from our informal definition of $t_1$, which we agree could be improved upon. We will update it with the following more precise definition: $t_1$ is the largest time smaller that $t_2$ such that $\|\|\lambda_{t_1}\|\| \ge \frac{c_1}{\rho}$ and $\|\|\lambda_{t_1-1}\|\|\le \frac{c_1}{\rho}$. Therefore, $\|\|\lambda_{t_1-1}\|\|\le \frac{c_1}{\rho}$ holds by definition. Remember that the Lagrangian multipliers are initialized to $0$, and at time $t_2$ they reach $\frac{c_2}{\rho}$, which is strictly larger than $\frac{c_1}{\rho}$.
>
> **On the paper by Ding et. al (2013):**
>
> We appreciate the reviewer for bringing this paper to our attention. We will include it in our discussion of related work on the stochastic BwK model. While the paper explores a variant of the stochastic BwK model, its connection to our work is relatively limited. Several key differences set the two settings apart: their setting assumes rewards and costs are generated i.i.d., whereas we focus on algorithms that remain robust even in adversarial environments; their costs are discretized; and their regret baseline is defined with respect to the stopping time of the algorithm rather than a fixed time horizon $T$. As a result, the guarantees they achieve (i.e., a regret bound of $O(\log B)$) are not comparable to those in our setting. In particular, in the stochastic case, our framework aligns with the standard lower bound of $\Omega(\sqrt{T})$ from the (unconstrained) multi-armed bandit problem, making a direct comparison challenging.

---

### Official Review · Reviewer_5ZmK · 2025-03-13

**Overall Recommendation:** 3

**Summary:**

This paper studies the general constrainted optimization problem where the reward and cost functions can either be stochastic or adversarial. By extending the LagrangeBwK framework by requiring the primal & dual algorithms to be weakly adaptive in addition to being no-regret, the authors designed a best-of-both-worlds algorithm which doesn't require the knowledge of the Slater's condition constant $\rho$. The results are exemplified for two constrained online learning problems.

**Claims And Evidence:**

Looks convincing, although I didn't verify the correctness of every claim

**Essential References Not Discussed:**

Didn't see any obvious missing references

**Experimental Designs Or Analyses:**

N/A

**Methods And Evaluation Criteria:**

Yes, they're consistent with previous ones in the literature

**Other Comments Or Suggestions:**

There's a typo in the citation of Lemma B.2.

**Other Strengths And Weaknesses:**

Strengths:

1. The baseline for adversarial case is stronger than previous ones. The upper bound is complemented with a lower bound, showing that the seemingly bad $1+\rho^{-1}$ competitive ratio is already the best one can hope if we do not want to incur $\Omega(T)$ violations. -- but see also Q1.
2. The idea why we need weakly adaptive properties in addition to no-regret is clearly depicted via Example 5.2.

Weakness:

1. The technical results are not explained, making it hard to understand why these technical results are of importance.
2. Because of 1, it is unclear what is the main technical contribution of this paper. It seems to me that everything is based on the observation that "a direct application of LagrangeBwK may result in super large dual variables; but if we additionally require the plug-in algorithms to be weakly adaptive, it works". See Q2.

**Questions For Authors:**

1. Regarding the Strength 1, is the worse competitive ratio due to the stronger baseline? That is, if the baseline is instead the standard one in the literature (which requires average cost <= ...), what's the best competitive ratio one can hope? If applicable, does the performance of your algorithm have a better guarantee in this case?
2. Is the understanding in W2 correct? What are the main technical contributions?

---

Both resolved in authors' rebuttals. Updated score accordingly.

**Relation To Broader Scientific Literature:**

Looks like the observation that weak adaptivity can result in bounded decision variables can be used in other constrained optimization problems.

**Theoretical Claims:**

I checked the proof of Lemma 6.2 Appendix B and it looks correct. I intuitively understood Proposition 5.3 via Figure 1. Didn't check Theorem 4.1 and those in Sections 7 & 8.

---

> ### Author Rebuttal · Authors · 2025-04-01
>
> ### **On the competitive ratio**
>
> Thank you for raising this possible source of confusion about the competitive ratio. The issue here is primarily one of nomenclature rather than our choice of a stronger benchmark. We agree that adding further clarification in the final version will be beneficial.
>
> The key distinction lies in the interpretation of $\rho$ in our work versus its meaning in the BwK literature. In our setting, we normalize the constraint so that costs must remain below a fixed threshold of 0, whereas in the BwK literature, the costs are constrained by a per-round budget of $\rho_{BwK}$.
>
> To see how our approach translates to the BwK model, consider how we would redefine the costs $g_t$ to solve BwK via our framework. Specifically, we could set $g_t(x) = \frac{x^\top c_t - \rho_{BwK}}{1 - \rho_{BwK}}$. Under this transformation, playing the void (``free’’) action in the BwK setting results in $g_t(0) = \frac{-\rho_{BwK}}{1 - \rho_{BwK}} = -\rho_{Adv}$, where $\rho_{Adv}$ is what we called $\rho$ in our paper. From this, it becomes evident that our competitive ratio of $1 + 1/\rho_{Adv}$ is equivalent to $1/\rho_{BwK}$.
>
> We thank the reviewer again for highlighting this potential source of confusion, and we will incorporate this clarification, including this short proof sketch, in the final version of the paper.
>
>
> ### **On technical contributions**
>
> Due to space constraints, we have deferred all proofs to the appendix and instead focused on providing a clear and intuitive explanation of the key ideas in the main paper. We are pleased that the reviewer appreciated our efforts to make these concepts more intuitive and accessible (especially how Example 5.2 highlights the necessity of using adaptive regret minimizers). However, we respectfully disagree with the idea that our contributions lack technical depth.
>
> Several key components of our proof are far from straightforward. In particular, in Section 6, we discuss the self-bounding lemma, which demonstrates that the Lagrangian variables remain automatically bounded when using scale-free primal algorithms. We find the proof of the main lemma (Lemma 6.2) highly nontrivial, and we will include a sketch of the proof in the main paper using the extra page available in the camera ready. See the answer to Reviewer Ldsh for more details on the proof (that we will include in the final version of the paper).
>
> Moreover, we believe that handling general constraints is a highly non-trivial task.Indeed, many previous works on BwK have attempted to achieve similar results under assumptions as weak as ours. However, they have only partially succeeded, addressing the problem only in the stochastic setting [1,2,3,4,5]. Removing the knowledge on the Slater parameter is both important on the practical size and interesting from a technical standpoint, even more so when it comes from the elegant idea of using adaptivity to self-bound the Lagrangian multipliers.
>
> These results are not only technically challenging but also highly relevant as they enable significant applications. In the paper, we highlight two key examples: bandits with general constraints and the contextual bandits with linear constraints problem recently introduced by Slivkins et al. (2023b). Both models have practical implications, including applications in autobidding for first-price auctions. Given the generality of our approach, it is likely that there are additional applications we have yet to explore.
>
> **References:**
>
> [1] Agrawal, Shipra, and Nikhil R. Devanur. "Bandits with concave rewards and convex knapsacks." Proceedings of the fifteenth ACM conference on Economics and computation. 2014.
>
> [2] Agrawal, Shipra, and Nikhil R. Devanur. "Bandits with global convex constraints and objective." Operations Research 67.5 (2019): 1486-1502.
>
> [3] Yu, Hao, Michael Neely, and Xiaohan Wei. "Online convex optimization with stochastic constraints." Advances in Neural Information Processing Systems 30 (2017).
>
> [4] Wei, Xiaohan, Hao Yu, and Michael J. Neely. "Online primal-dual mirror descent under stochastic constraints." Proceedings of the ACM on Measurement and Analysis of Computing Systems 4.2 (2020): 1-36.
>
> [5] Castiglioni, Matteo, et al. "A unifying framework for online optimization with long-term constraints." Advances in Neural Information Processing Systems 35 (2022): 33589-33602.

---

> > ### Comment · Reviewer_5ZmK · 2025-04-01
> >
> > Thank you for your response to my Q1. Yes, the claim of "stronger result" definitely makes more sense if the two competitive ratios are in fact identical.
> >
> > Thanks for listing the technical contributions. When I looked at Appendix B I felt it was fairly standard, for example the following paper (which studies a completely unrelated problem) also uses a similar argument that "since $\eta$ is small, it takes many rounds for a coordinate to be large".
> >
> > Zihan Zhang, Wenhao Zhan, Yuxin Chen, Simon S Du, Jason D Lee. "Optimal Multi-Distribution Learning". COLT 2024.
> >
> > But now I realize they are actually pretty different, because in the above paper they are studying something over a simplex, but here the dual variables have unbounded regions. I agree that the results regarding almost-independent coordinates are far from intuitive. I encourage the authors to include more discussions regarding the technical difficulties in the revision.
> >
> > I am now leaning more towards accept.

---

> > > ### Author Response · Authors · 2025-04-02
> > >
> > > Thank you for taking the time to read our responses. We appreciate your reassessment of the contributions of our paper, and hope this can be reflected in your final score.

---

### Decision · Program_Chairs · 2025-05-01

**Decision:**

Accept (poster)

**Comment:**

This paper extending the bandits with knapsacks (BwK) framework to the setting with long-term constraints. By exploring weakly adaptive primal and dual algorithms, the authors develop best-of-both-worlds guarantees for stochastic and adversarial inputs, without knowing the  Slater’s parameter. Given the strength of the contribution, all reviewers unanimously recommend acceptance. That said, the authors are encouraged to further improve the paper, particularly in terms of paper presentation.